# Preliminary evaluation of the potential of tree-ring cellulose content as a novel supplementary proxy in dendroclimatology

Malin M. Ziehmer[1,2], Kurt Nicolussi[3], Christian Schlüchter[4,2], Markus Leuenberger[1,2]

[1]Climate and Environmental Physics, Physics Institute, University of Bern, 3012 Bern, Switzerland
[2]Oeschger Center for Climate Change Research, University of Bern, 3012 Bern, Switzerland
[3]Institute of Geography, University of Innsbruck, 6020 Innsbruck, Austria
[4]Institute of Geological Sciences, University of Bern, 3012 Bern, Switzerland

*Correspondence to*: Malin M. Ziehmer (ziehmer@climate.unibe.ch)

**Abstract.** Cellulose content (CC [%]) in tree rings is usually utilized as a tool to control the quality of the α-cellulose extraction from tree rings in the preparation of stable isotope analysis in wooden tissues. Reported amounts of CC [%] are often limited to mean values per tree. For the first time, CC [%] series from two high Alpine species, *Larix decidua* Mill. (European Larch, LADE) and *Pinus cembra* L. (Swiss stone pine, PICE) are investigated in modern wood samples and Holocene wood remains from the Early and Mid-Holocene. Modern
CC [%] series reveal a species-specific low-frequency trend independent from their sampling site over the past 150 years. Climate-cellulose relationships illustrate the ability of CC [%] to record temperature in both species, but for slightly different periods within the growing season.

The Holocene CC [%] series illustrate diverging low-frequency trends in both species, independent of sampling site characteristics (latitude, longitude and elevation). Moreover, potential age trends are not apparent in the two
coniferous species. The arithmetic mean of CC [%] series in the Early and Mid-Holocene indicate low CC [%] succeeding cold events. In conclusion, CC [%] in tree rings show high potential to be established as novel supplementary proxy in dendroclimatology.

## 1 Introduction

Highly-resolved proxy records from the tree-ring archive contribute significantly in reconstructing and
25 understanding past climate variability on versatile temporal and spatial scales (Büntgen et al., 2011; Esper et al., 2002; Mann et al., 1998). The annually-resolved and calendar-dated tree-ring records are particularly useful in disentangling natural climate variability from the anthropogenic imprint on the global climate and its impact on the climate system, which enables the improvement of models in view of projecting future climate change (Keel et al., 2016; Keller et al., 2016).

Most dendroclimatological reconstructions are based on tree-ring width (TRW) and maximum latewood density (MXD) records (Büntgen et al., 2006; Ljungqvist et al., 2016; Trouet et al., 2009). However, the determination of stable isotope ratios of carbon ($\delta^{13}$C), oxygen ($\delta^{18}$O) and hydrogen ($\delta$D) isotopes in tree components and their ability to reconstruct past climate has gradually opened a new field of dendroclimatology during the past four decades (Frank et al., 2015; Leuenberger, 1998; McCarroll and Loader, 2004; Treydte et al., 2006). In addition,
the potential of stable isotope records in tree rings has been increased by improvements in both sample preparation and measurement techniques (Boettger et al., 2007; Filot et al., 2006; Laumer et al., 2009; Loader et al., 2015). Even though the isotopic composition has been determined in various components of a tree, e.g. bulk

wood samples, leaves, methoxyl groups of lignin and leaf waxes (Anhäuser et al., 2017; Borella et al., 1999; Kahmen et al., 2011; Kimak et al., 2015), the most preferred component is α-cellulose extracted from tree rings (Borella et al., 1998, 1999, Loader et al., 2003, 2013; McCarroll and Loader, 2004; Treydte et al., 2007). A major advantage of using α-cellulose in isotope research is the extraction of one single chemical component of a

tree ring. Thereby it allows researchers to circumvent the biases that potentially could result from a changing composition (i.e. in the cellulose/lignin ratio) when analysing bulk wood since different components exhibit significantly different isotope compositions (Borella et al., 1998, 1999).

The α-cellulose is the dominant component in tree rings and its extraction follows a standardized procedure (Boettger et al., 2007). The woody annual growth layers of trees, i.e. the tree rings, are composed of lignin

(phenolic polymer), hemicellulose (heterogenous polysaccharides) and cellulose (β-1,4 glucan) in a ratio of 1:1:2 (Freudenberg, 1965; Hu et al., 1999). Thereof, the part of cellulosic material which is insoluble in a 17% sodium hydroxide (NaOH) solution is defined as α-cellulose (Burton and Rasch, 1931; Cross and Bevan, 1912). The various tree-ring components are known to differ in their isotopic composition, whereas α-cellulose reveals to be the most stable component as the polymer is of long-term stability (Boettger et al., 2007; Leavitt and Danzer,

1993; McCarroll and Loader, 2004).

The quantity of the α-cellulose content (hereinafter referred to as CC [%]), generally calculated from the dry weight and the α-cellulose weight of a wood sample, is usually determined and used as a tool to test the quality of the cellulose extraction and the purity of the cellulose (Burton and Rasch, 1931).

However, variations in the amount of α-cellulose and the determined CC [%] (also: cellulose yield; used rather

simultaneously in literature) are scarcely mentioned in dendroclimatological literature. Merely few publications provide numbers on the mean extracted CC [%] (Cullen and Macfarlane, 2005; Gaudinski et al., 2005; Leavitt and Danzer, 1993; Loader et al., 1997). While reported results on the CC [%] are scarce in literature, the importance of the CC [%] calculation as a tool for the determination of degradation state in subfossil wood is addressed within several studies (Brenninkmeijer, 1983; Loader et al., 2003 and references therein; Schleser et

al., 1999).

Schleser et al. (1999) point out that the determination of isotopic ratios in cellulose from woody plant tissues often assumes constancy in the CC [%], which may hold true for modern samples from living trees. However, samples from subfossil or fossil wood have presumably experienced degradation processes with potential influence on the CC [%]. The artificial degradation of samples revealed a discrimination of $\delta^{13}C$ values up to

0.3‰ with decreasing CC [%], where the change in degradation behaviour over time splits into initially increased degradation of cellulose with a strong depletion in $\delta^{13}C$, followed by an inverse effect comprising an enrichment in $\delta^{13}C$ with advancing degradation (Schleser et al., 1999).

Loader et al. (2003) discusses the use of whole wood versus α-cellulose in subfossil tree samples as the various wood components (lignin, hemicelluloses, cellulose) experience variable degrees of decomposition changing the

ratio of the individual components within a tree ring. The polysaccharide components, i.e. cellulose and hemicelluloses, decompose more rapidly than lignin, which leads to an altered ratio of cellulose and lignin in whole wood samples which in turn modifies the isotopic composition of whole wood samples (Borella et al., 1998; Loader et al., 2003; Schleser et al., 1999). Therefore, cellulose is the preferred wood component in subfossil wood studies; however, the effect of partial decay of cellulose on the isotopic signature of tree rings

and its implications for paleoclimate reconstructions is insufficiently investigated, although results from

naturally degraded subfossil wood show no evidence for a change in the isotopic composition of cellulose due to degradation (Loader et al., 2003).

While studies on CC [%] and its implication for dendroclimatological studies are scarce, even less is known on the influence of environmental conditions on the annual variability of the CC [%] (Genet et al., 2011). For instance, Gindl et al. (2000) discussed lignin content as a temperature proxy for the late growing season (Sept-Oct). As lignin and cellulose are among the major components of a tree ring, and changes in one component will affect the cellulose/lignin ratio, a link between CC [%] and temperature can be expected as well. However, the study of Gindl et al. (2000) covers only a short time period of 10 years and the method used is rather time-consuming, whereas the production of CC [%] is often a by-product during the extraction of α-cellulose for stable isotope analysis in tree rings. A study on CC [%] in roots revealed influences of topography and soil moisture status on root CC [%] (Hales et al., 2009). However, variations in CC [%] over time may as well be related to tree metabolism and the availability in non-structural carbon (NSC), which occurs in the form of starch and sugars, where the sugars provide the nutrients for cell wall components such as cellulose (Haigler et al., 2001; Hoch et al., 2002). Studies on NSC concentrations at the Alpine tree-line provide evidence that trees growing in the tree-line ecotone are not depleted in carbon; on the contrary, NSC concentrations are found to increase with elevation, implying that the limitation of tree growth at the upper tree-line does not result from insufficient nutrition, but rather thermal conditions, i.e. temperature, which limits the sink activity and thereby growth (Hoch et al., 2002; Hoch and Körner, 2003, 2009, 2012; Körner, 2003). Applied to the CC [%] of a tree ring, it can be hypothesized that temperature acts as the main control, as it is steering the sink activity, i.e. the tissue formation.

This manuscript presents results of a pilot study, where time series of modern and subfossil tree-ring cellulose are established, allowing a novel insight into a tree-ring component which is commonly used in dendroclimatology, but hardly examined itself in its variability over time and in the factors driving its variability. The current study is embedded in the project *Alpine Holocene Tree Ring Isotope Records* (AHTRIR) which aims at reconstructing Holocene climate variability using a multi-proxy approach for the past 9000 years. Therefore, tree-ring material consists only partly of living wood material; the by far largest part is based on findings of Holocene wood remains from glacier forefields, peat bogs and small lakes in the central European Alps. Initially, the determination of CC [%] was used to determine the quality of the cellulose extraction before performing analysis of triple isotopes ($\delta^2$H, $\delta^{18}$O, $\delta^{13}$C) on the tree-ring cellulose. However, it offers the unique opportunity to investigate the CC [%] and its variations in long-living trees from two high-Alpine coniferous tree species (*Larix decidua* Mill. and *Pinus cembra* L.) in Holocene wood remains found in glacier forefields and peat bogs in the European Alps as well as in modern trees sampled at the current Alpine tree line. The case study aims at (i) identifying common trends in time series of CC [%] in tree rings from Early- and Mid-Holocene subfossil and modern wood material; (ii) investigating dependencies of CC [%] in relation to their sampling site (latitude, longitude, elevation), their age and the species; (iii) analysing climate-CC [%] relationships and finally (iv) determining the ability to utilize CC [%] as a supplementary proxy in dendroclimatological reconstructions.

## 2 Material and methods

### 2.1 Samples and sampling sites

Holocene wood remains and modern tree-ring material are sampled with the intent to reconstruct Holocene climate variability in a multi-proxy approach. Thereby, samples from two Alpine tree-line species, namely the Swiss stone pine (*Pinus cembra* L., PICE) and the European larch (*Larix decidua* Mill., LADE) are collected as they represent the typical tree-line species present at modern sampling sites in form of living trees, and in glacier forefields and peat bogs as subfossil wood remains (Joerin et al., 2008; Nicolussi, et al., 2009; Nicolussi and Patzelt, 2000a). The sampling sites are located along a SW-NE transect in the Central European Alps (46°02'N - 47°03'N, 7°33'E - 12°15' E; Fig. 1, Table 1 and 2). The modern tree-ring material has been cored at three upper tree-line sites close to glacier forefield sites during fieldwork in 2015 (Figure 2, Table 1). At each site, four trees per species have been cored; however, at the sites UAZR and FPCR, only one of the two species is present. Cores are taken at breast height from three radial sections; two in parallel to the slope, and one up-slope, thereby avoiding any kind of compression wood on the down-slope side of the tree.

The Holocene wood remains stem from glacier forefields, peat bogs and small lakes, which have been continuously collected over the last two decades (Joerin et al., 2006, 2008; Nicolussi, et al., 2009; Nicolussi et al., 2005; Nicolussi and Patzelt, 2000b). Wood material of the EACC (Eastern Alpine Conifer Chronology) has been merged and updated with subfossil samples of the same species and altitude collected by continued sampling of wood remains and stem discs in glacier forefields (Nicolussi and Schlüchter, 2012). Similar to modern wood samples, the use of compression wood in these Holocene samples was avoided during the further analysis.

### 2.2 Tree ring width and dating

Total tree-ring width (TRW) is measured for all samples. At least two radii are measured per tree with a precision of 0.001 mm by using a LINTAB device in connection to TSAPWin software (Rinn, 1996). For each sample, a mean TRW series is established by crossdating and averaging the radii per tree. In case the inner part of the stem is missing in a tree-ring series, the tree-ring number between the pith and the first tree ring measured is estimated (pith-offset (PO), Table 3). For the modern series, dating is simple as the exact sampling date is known. For the Holocene wood remains, the mean series are compared to local tree-ring series and reference chronologies (Nicolussi, et al., 2009; Nicolussi and Schlüchter, 2012). Crossdating is achieved by calculating statistical parameters, and applying visual controls.

### 2.3 Sample preparation

Samples for CC [%] analysis are prepared as 5-year tree-ring blocks per tree, thus no pooling between trees is applied. For the Holocene samples, radial wedges are cut out of stem disks by using a bandsaw. Subsequently, blocks are prepared from the wedges and the cores under the microscope by using a commercial scalpel. The dry weight of each block is scaled and is defined to be between 20 mg and 50 mg, as a 60 % weight loss is expected during α-cellulose extraction. Further, the incipient degradation in samples might have additionally reduced the CC [%] in the samples. The blocks are further cut into smallest slices of wood material by first slicing in parallel to the fibre direction, and subsequently against the fibre direction. The resulting size of the cut sample is < 1

mm. Again, the dry weight of the cut sample is scaled to estimate the loss during the cutting procedure. For the following chemical treatment, the samples are packed into ANKOM Filter Bags (Type F57), heat-sealed and labelled individually.

**2.4 Cellulose extraction**

5     The cellulose extraction is based on a modified Jayme-Wise procedure (Boettger et al., 2007; Borella et al., 1999; Leuenberger, 1998). The bleaching (delignification) of samples in a 1 % $NaClO_2$ + $CH_3COOH$ solution with a pH value of 4 is conducted at 70 °C for at least 30 hours to remove lignin. In a second step, samples are treated in a 17 % NaOH solution at 25 °C for 45 minutes to extract the pure α-cellulose (Burton and Rasch, 1931; Cross and Bevan, 1912). Subsequently, samples are neutralized in a 1 % HCl solution using 25 % HCl and 10   washed intensely with heated and cold distilled water until the pH of the washing water is ~ 7. Samples are dried in an oven at 50 °C; the duration of the drying is thereby dependent on the sample size in the individual filter bags.

**2.5 Calculation of CC [%]**

The CC [%] is calculated from the dry weight of wood and the α-cellulose weight of a sample:

$$CC \ [\%] = \left(\frac{\alpha - cellulose \ weight}{wood \ dry \ weight}\right) \times 100\% \tag{1}$$

The wood dry weight is thereby defined as the sample weight of the individual wood sample after cutting, and the cellulose weight refers to the weight of the extracted α-cellulose after being removed from the ANKOM filter bag.

A remaining source of error is the collection of dry α-cellulose from the ANKOM filter bags. The described slicing of wood samples facilitates the removal of α-cellulose from filter bags compared to ground wood samples and thereby reduces the loss of α-cellulose material when being collected from the filter bags. In this study, the resulting loss is examined for 42 samples, where the filter bag including the α-cellulose and the emptied filter bag are scaled in addition to the α-cellulose weight to give an accurate estimate of the observed loss.

**2.6 Outlier detection and correction**

Outliers in CC [%] are observed for Holocene wood remains, where the outermost rings have experienced a higher degree of degradation, e.g. due to abrasion at the glacier sites or the exposition of wood to environmental influences and associated weathering of the sample. The differential grades of degradation lead to pronounced decreases in CC [%], appearing within the outermost rings, but also along cracks between individual rings and 30   seldom on the innermost parts of trees, which complicate the comparison of individual time series. Therefore, detection and elimination of outliers is performed by visual and statistical application of boxplots (Tukey, 1977). Tukey (1977) defines outliers as samples, which lie outside the whiskers, whereby the extension of the whisker is limited by the 1,5 inner-quartile range (IQR). In addition to the boxplots, visual comparisons of suspicious samples with other cellulose series is applied; thereby outliers close to the whiskers revealed to be true values 35   and the loss of samples by outlier detection and elimination could be minimized to the most extreme values.

**2.7 Meteorological data**

CC [%] series of modern sampling sites are correlated with meteorological data from the HISTALP database to investigate the relationship between climate and CC [%] (Auer et al., 2007). Data is obtained in the form of Coarse Resolution Subregional Means (CRSM), which are calculated as arithmetic means from homogenized

individual station series for five defined subregions within the Greater Alpine Region (GAR), which have been identified by EOF-based regionalization. Auer et al. (2007) recommend CRSM-series especially for lower frequency analysis for all climate parameters. Series from the sampling sites FPCR and UAZR are correlated to meteorological data from the NW subregion, and from VRR to data from the SW region, respectively, due to their geographical location.

For these subregions, monthly anomalies (calculated for the reference period 1961-1990) of temperature (T [°C], 1760-2008), precipitation (prec [mm], 1800-2008), cloud cover (cloud [%], 1840-2008) and sunshine (sun [h], 1880-2008), including the arithmetic seasonal (MAM, JJA, SON, DJF), semi-annual (AMJJAS, ONDJFM) and annual means are extracted. For further analysis, 5-year mean values are calculated for the annually resolved climate data as CC [%] series were established in 5-year resolution and which thereby integrate the mean

climatic signal over five consecutive years.

Pearson's correlation coefficient is calculated between the mean CC [%] series per site and species, and the monthly, seasonal and annual values for the four climate variables for the time period 1865-2008 (1880-2008 in case of sunshine). We assume here that each data point (5-year blocked data) in the climate variable datasets and the CC [%] series is independent, since there is no data point overlap.

**3 Results**

As the precise calculation of the CC [%] depends on error estimates for the dry weight of samples and the cellulose weight, two steps in the calculation of CC [%] are here improved: the precision of dry weight and the loss estimation in cellulose weight.

The mean sample loss during the cutting process amounts to 2.6 ± 1.7 % of the dry weight of an individual

sample, where mean loss values range from 0.3 % up to 11.1 % per tree (data not shown here). Besides, very few individual samples experienced losses between 10 % and 30 % caused by wood pieces bouncing off during cutting or loss of material due to powdery wood substance as a result of degradation. Hence, for the precise calculation of CC [%], a determination of sample weight after cutting or milling is essential.

Second, when unpacking the cellulose material from the filter bags, there is always a risk that smallest fibres

remain in the filter bag or fly off during the removal; therefore, we assume this error to be systematic. In this study, we estimated this error by analysing 42 filter bags, which revealed a mean loss of α-cellulose of 0.378 ± 0.163 mg (3.2 ± 1.4 %) (Table 4). Therefore, the calculation of the CC [%] results in the minimum CC [%] of the sample, as losses of cellulose in the order of 3.2 ± 1.4 % of the cellulose weight are to be expected, which would result in slightly increased CC [%] values. However, as the unpacking is accomplished equally for all samples, a

systematic error is assumed which affects the CC [%] calculation in the same manner. It can be assumed that the systematic error does not influence the variability of the individual CC [%] series.

Further, the relative uncertainty in cellulose weight transfers directly to the relative uncertainty in CC [%] determination as relative uncertainties are additive, with the uncertainty of the dry weight after cutting being

negligible. Relevant for the CC [%] variation is the variation of the relative uncertainty (±1.4 %), and not the relative uncertainty itself (3.2 %), which only yields a mean offset of the whole curve. Although the relative uncertainty slightly limits the interpretation of minor differences between the individual 5-year CC [%] samples, it does not limit the investigation of trends in CC [%] series.

The improvement in the CC [%] calculation and the estimation of the relative error justify the use of the term "CC [%]" rather than "cellulose yield", as sources of error are reduced and well estimated, and resulting calculations are close to the true values.

**3.1 Modern CC [%] series**

For a better understanding of CC [%] variations, modern CC [%] series are analysed for their temporal

variability per species and site, where the absolute values and variations in CC [%] are given in the unit CC%. Trees sampled at UAZR consist solely of PICE samples, as the tree-line at that site is only composed of one tree species. Measured tree-ring width series vary in length between 150 and 205 years; the pith offset estimation varies between 32 and 82 for the individual trees (Table 3). The mean segment length (MSL) amounts to 173 years and the period covered by all four samples spans from 1860 to 2014 A.D. (Table 1). The mean cellulose

content per tree varies between 31.6 CC% and 39.7 CC%, where UAZR-1 (33.2 CC%) and UAZR-2 (33.2 CC%) show distinctly lower mean CC [%] values as UAZR-3 (38.6 CC%) and UAZR-4 (39.7 CC%). These differences are a result of low minimum values for UAZR-1 and UAZR-2, which are up to 10 CC% lower than for the other two trees (Table 5). Further, the time series of CC [%] for UAZR -1 and -2 show more variability in their series compared to UAZR-3 and -4, which show a high common variability and merely small fluctuation in

their CC [%]. This is further confirmed by the calculated Pearson's correlation coefficient (r) with highest values for UAZR-3 and -4 (r=0.64) (Table 6). A significant correlation is also found between UAZR-2 and -3 (r=0.36), whereas the remaining correlation coefficients are non-significant ($p > 0.05$). The arithmetic mean calculated from four samples in their common period (1860-2015 AD) indicates an increase in CC [%] over time by ~ 5 CC% (red line, Fig. 3, top left).

In contrast to UAZR, the sampling site FPCR consists solely of LADE trees, where tree-ring series contain between 155 and 335 years; the MSL amounts to 208 years, raised by the sample FPCR-1 (335 years), whereas the residual tree-ring series consist of 155 and 170 years (Table 1, Table 3). The four tree samples cover a common period from 1860 to 2015 A.D. (Table 1). The mean CC [%] per tree varies between 28.3 CC% and 32.9 CC%, showing overall lower mean CC [%] values than observed for the PICE trees at UAZR (Table 5).

Further, the range of CC [%] per individual tree is large, as LADE trees show low minimum CC [%] values between 15.2 CC% and 21.8 CC% as well as high maximum values of up to 51.6 CC% (Table 5). In their common period, trees at FPCR show a decrease in CC [%] from 1860 to 1945 A.D. and a strong increase of nearly 20 CC% from 1980 to 2014 A.D, which is common among the individual trees (Fig. 3, top right). Pearson's correlation coefficient illustrates the common variability among the samples FPCR-1, -5 and -6,

showing high correlation coefficient between 0.67 and 0.79, whereas non-significant and lower correlation coefficients are found for FPCR-3 (Table 6).

The sampling site VRR located in the Val Roseg in Eastern Switzerland, and in the proximity of the Tschierva glacier, is the only sampling site where both tree species, LADE and PICE are found, thereby allowing a direct comparison of species under same climatic growing conditions.

PICE trees at VRR exhibit tree-ring series of 170 to 255 years, with a common period between 1845 and 2015 A.D. and a MSL of 209 years (Table 1, Table 3). Mean CC [%] varies between 32.9 CC% and 37.2 CC%, where the samples VRR-1 and VRR-1-2 (same tree) reveal a range of > 20CC%, whereas the other PICE trees at VRR exhibit sizes between 13 CC% and 16.3 CC% (Table 5). Minimum values are around 20 CC% for VRR-1 (and

1-2) and around 30 CC% for all the other trees, and maximum values exceed 40 CC% with maximum values > 45 CC% (Table 5). Calculated correlation coefficients are almost all significant ($p < 0.05$) and are between 0.41 and 0.75 (Table 6). The arithmetic mean series for VRR-PICE displays an increase in mean CC [%] of > 10 CC% in the common period (Fig. 3, bottom left).

The LADE tree-ring series at the same location, with a common growth period between 1865 and 2015 A.D. and

MSL of 219 years, consist of two times 150 years, 190 and 385 years (Table 1, Table 3). Mean CC [%] is between 26.3 CC% and 36.0 CC% with minimum values ranging from 16.3CC% to 30.4 CC%, whereas the maximum values per tree show a lower spread between 39.4 CC% and 46.3 CC% (Table 5). As for PICE, LADE CC [%] series reveal significant correlation coefficients among the individual trees (Table 6). The arithmetic mean series shows a decrease in CC [%] of approximately 5CC% in the period 1865-1980 A.D., followed by a

strong increase of more than 10 CC% during the recent three decades (Fig. 3, bottom right).

The investigation of individual tree-ring CC [%] series from three sampling sites (UAZR, FPCR, VRR) and two coniferous tree species (PICE, LADE) display common variability of trees from the same species, independent from the sampling site. Individual CC [%] per site and species show in general good agreement, represented in their correlation coefficients, which allows the establishment of arithmetic mean chronologies for their common

growth periods (comparable time periods for all sites).

Common trends in the tree species, independent of the sampling site, indicate a common influence from environmental factors on CC [%]. All PICE series reveal a continuous increase of CC [%] over time, whereas the LADE samples show a decrease over the common period, replaced by a rapid increase over the past three decades (Fig. 4, left). The close relationship between the site series of the same tree species is also revealed by

the calculated Pearson's correlation coefficient, where r=0.86 for LADE and r=0.76 for PICE mean series.

**3.2 Climate-cellulose relationships**

To test the influence of environmental conditions on the CC [%], Pearson's correlation coefficient is calculated between the mean CC [%] series per site and species and the climate variables temperature (cf. Fig 4, right), precipitation, sunshine as well as cloud cover (Fig. 5-6, Fig. S1-2).

A temperature signal is clearly recorded in both tree species, independent from the sampling site (Fig. 5). Both PICE sites show significant correlations with temperature throughout the year, with high correlation during the growing season and interestingly in October (r=0.67 for UAZR-PICE, r=0.65 for VRR-PICE). Seasonal temperature averages result in highest correlations for PICE; correlations for the summer season (JJA) result in r=0.68-0.71 and for autumn season (SON) in r=0.72-0.75 for UAZR-PICE and VRR-PICE, respectively. In

LADE, correlations with temperature are lower than for PICE and significant correlations are found mostly during the growing season (May-August). Seasonal averages in temperature anomalies result in highest correlations for the summer season (JJA) for both FPCR-LADE (r=0.55) and VRR-LADE (r=0.69).

Precipitation is best recorded in PICE at UAZR during the early growing season, revealing highest correlation coefficients in May (r=0.68). In contrast, LADE mean series at both sites and PICE at VRR mostly show non-

significant correlations with precipitation records (Fig. 6).

The influence of cloud cover is highest in PICE trees during the early as well as towards the end of the growing season (Fig. S1). Significant positive correlations are found in MAM for UAZR-PICE (r=0.49), and for JJA in both UAZR-PICE (r=0.41) and VRR-PICE (r=0.42). Interestingly, the correlation coefficient for cloud cover and LADE mean series reveals significant negative correlations during the winter season (DJF, r=-0.56 (FPCR-
LADE), r=-0.63 (VRR-LADE)), especially in January for FPCR-LADE (r=-0.57) and in February for VRR-LADE (r=-0.46).

High, significant correlations with sunshine are observed for the late autumn and winter season, whereas negative correlations are found during summer season, being significant at both PICE sites in August, where VRR-PICE reveals significant negative correlations with sunshine from May to August (Fig. S2). For UAZR-
PICE, highest correlation with sunshine are found in January (r=0.86) as well as in February (r=0.76) and the winter season (DJF, r=0.87), whereas for VRR-PICE, highest correlations are found for November and December (r=0.84 and r=0.82, respectively) as well as for the winter season (r=0.79). The pattern is similar for FPCR; significant positive correlations occur in late autumn and winter (N, D, J, F) and are highest in January (r=0.62). In contrast, VRR-LADE shows highest correlations in late autumn (Nov) and in February (r=0.66 and
r=0.55, respectively).

In summary, the results of the Pearson's correlation coefficient analysis between HISTALP data and mean CC [%] per species and site indicate partly significant influence of environmental factors (temperature, precipitation, cloud cover, sunshine duration) on the CC [%] in tree rings and the potential to reconstruct past climate from this novel supplementary proxy. However, the comparison of the two coniferous species further indicates a species-
specific response of CC [%] on the environmental conditions.

**3.3 Holocene CC [%] series**

In a further step, CC [%] series are established for LADE and PICE tree-ring samples from glacier forefields, peat bogs and small lakes for the period from 8550 to 3500 years b2k (Fig. 7). To investigate trends in the CC [%], the individual series are averaged per species (PICE in blue, LADE in green) and an arithmetic mean is also
calculated over all series (black), which is smoothed by a spline (orange) to illustrate long-term changes (Fig. 7). The arithmetic mean series shows the high variability in the CC [%] over time, fluctuating between 22 CC% and 40 CC%. Further, the series exhibits interesting low-frequency trends with rapidly decreasing CC [%] in the periods 8250-7950 years b2k ($\Delta$ CC [%] ~ 5 CC%), 6250-5950 years b2k ($\Delta$ CC [%] ~ 2 CC%), 5650-5450 years b2k ($\Delta$ CC [%] ~ 4 CC%), 5300-5000 years b2k ($\Delta$ CC [%] ~ 5.5 CC%) and 4500-4000 years b2k ($\Delta$ CC
[%] ~ 6 CC%). Besides these phases of rapid decreases, CC [%] is increasing on a multi-centennial scale between 7350 and 6250 years b2k ($\Delta$ CC [%] ~ 4.5 CC%) and more rapidly after low CC [%] phases, e.g. in the periods 7950-7750 years b2k ($\Delta$ CC [%] ~ 3.5 CC%), 5450-5300 years b2k ($\Delta$ CC [%] ~ 7 CC%) and 4000-3650 years b2k ($\Delta$ CC [%] ~ 8.5 CC%). The calculated arithmetic mean series per species exhibit mostly lower mean values in the LADE series (green) and more positive values for PICE (blue). LADE shows extremely low CC
[%] between 4700 and 4200 years b2k, and extreme high values between 4000 and 3900 years b2k. The PICE average series shows extremely low values around 8000 years b2k as well as around 4000 years b2k; otherwise both the LADE and PICE series mostly fluctuate between 30 CC% and 40 CC%.

For most of the investigated time period, the sample replication amounts to 3 trees; two gaps at 7300 and 6300 years b2k are solely covered by a single tree; partly a replication of $\geq$ 4 trees is achieved (Fig. 7, bottom).

Fig. 7 further displays identified cold phases (light-blue bars) analysed by Wanner et al. (2011), where the darker bars illustrate the cold phases which were identified by three different phenomena (cold phases, glacier advances, Bond cycles) and the lighter areas mark the entire length of the cold phases; additional cold periods are marked as blue vertical lines (Wanner et al., 2011, 2015). Interestingly, the strong decreases in cellulose content emerge mostly after the indicated cold phases.

As modern wood samples already displayed differential trends in the CC [%], mean series of LADE and PICE and their low-frequency trends are investigated closer in Fig. S3, where the contribution of individual trees per species is revealed in the sample replication (Fig. S3 (d)). The mean series reveal distinct offsets over time (Fig. S3 (a)). In the periods 8100-7800 years b2k and 5600-4600 years b2k, the smoothed values exhibit a common long-term trend, whereas in the remnant phases, series even show opposed trends. The difference between the two arithmetic mean series emphasizes that the differences between the species are not constant over time (Fig. S3 (b)).

Although the Holocene samples stem from various sites, the phases 8900-8200 years b2k and 7500-6700 years b2k are dominated by glacier forefield sites at Mont Miné and Tschierva glacier, respectively (Fig. S3 (c)). For both intervals, the two coniferous species are present, and as the sampling site is equal, similar as well as divergent trends in the species result presumably from the impact of environmental factors on CC [%].

In order to confirm the non-existing influence of the sampling site location on CC [%] series also in Holocene samples, the arithmetic mean of CC [%] per site and species is calculated and compared to latitude, longitude and elevation of the individual sites (Fig. S4). Calculated linear regressions (dotted lines) per species do not indicate a dependence of CC [%] and sampling site locations, as linear regressions are non-significant ($p > 0.05$). To further exclude non-climatic influences on CC [%], individual CC [%] series per tree are aligned according to their biological age by taking the estimated pith-offset into account (Fig. S5). Both PICE (upper panel) and LADE (lower panel) do not reveal any distinct low-frequency trend in the age-aligned series.

## 4 Discussion

The determination of CC [%] is a common procedure in tree-ring laboratories that measure stable isotope ratios in tree-ring cellulose. However, to our current knowledge, existing records of CC [%] were simply used as a tool to ascertain the quality of the extracted cellulose; the CC [%] is, if mentioned in literature, given as a mean value and its temporal variability is not regarded any further.

The investigation of CC [%] in modern tree samples of two high-Alpine coniferous species (LADE, PICE) at three sites in the Swiss Alps reveals common variations within the tree species, independent of the sampling site, pointing out a common environmental driver. While PICE trees show a continuous increase over the investigated common period, LADE trees are characterized by a decrease throughout the 20th century followed by a strong increase in CC [%] from the 1980s until today.

In order to exclude any age-related biases, possible age trends in CC [%] series have been investigated, even though they were not expected as the trends in modern CC [%] series of the two tree species diverge and do not reveal a common increase/ decrease over time. The age-alignment of both modern and Holocene CC [%] series by their biological age, taking their estimated pith-offset (PO) into account, illustrates that CC [%] series are not biased by age trends, which leads us to the conclusion that the trends which we find in CC [%] series are most probably driven by climatic variables (Fig. S5).

Further, the two different species *Larix decidua* Mill. and *Pinus cembra* L. at the sampling site Val Roseg (VRR) exhibit different trends in their mean chronologies. As they experience the same climate, the role of the biological factors here is undoubted. This is rather obvious as two coniferous species are compared, where *Larix decidua* Mill. is a deciduous species, whereas *Pinus cembra* L. is an evergreen species; therefore, differences in

their metabolism are to be expected. Although the two species are both found at the upper tree-line and known to be adapted to the harsh environmental conditions, LADE is characterized as the light-demanding pioneer species which is often found in open settings, e.g. on glacier forefields, whereas PICE is, under undisturbed conditions, the highest rising species in the inner sections of the Alps and with that adapted to short vegetation periods (Ellenberg, 1996). The fact that individual CC [%] series from the same species at different sites are similar calls

for a common driving factor of regional extent, such as temperature.

The investigation of the climate-cellulose relationships by correlating mean CC [%] series with climate variables (temperature, precipitation, cloud cover, sunshine duration) extracted from the HISTALP database (Auer et al., 2007) over a common period of 140 years (1865-2008 A.D.; 1880-2008 for sunshine) reveals interesting correlation coefficients between CC [%] and climate. Temperature is recorded in both species, revealing highest

correlations during summer season, and additionally for the autumn season in PICE trees. An increase in CC [%] in trees with increasing temperatures may be explained by the enhanced sink activity in trees at the upper tree-line. In the attempt to establish a functional explanation of the Alpine tree-line, Körner (1998) compiled five hypotheses, where the climate-driven tree-line is either driven by stress, disturbance, reproduction, carbon balance or growth limitation (Körner, 1998). Since then, several publications have shown that concentrations of

non-structural carbon (NSC) in trees at the upper tree-line increase with elevation, thereby indicating no evidence that trees at high elevations experience a carbon shortage, but rather implying a lowered sink activity at the upper tree-line resulting in a growth-limitation (Hoch et al., 2002; Hoch and Körner, 2009, 2012; Körner, 2003). As photosynthetic activity is less sensitive to cold temperatures than growth, it may still be active at 0 °C, where temperatures are too cold for a tree to grow (minimum temperatures for tree growth are ~ 5-7 °C) (Hoch

and Körner, 2012; Körner, 1998; Tranquilini, 1979). Therefore, the carbon acquisition in a tree can still be active, although low temperatures prevent the tissue formation. In consequence, the concentration of NSC in trees at the upper tree-line is increasing, as the source is still active, but the sink is not. In conclusion, colder temperatures during the growing season reduce the tissue formation in tree rings, resulting in less CC [%] in the tree ring as the source activity is low. Regarding the investigated species in this study, the LADE as a pioneer

species, exploring the upper tree-line, highly benefits from the increase in temperature associated with global warming and which is especially pronounced since the 1980s (Rebetez and Reinhard, 2008). As temperatures during the growing season increase and the growing season is potentially prolonged, the sink activity and the CC [%] in tree rings are increased. This might explain the found correlations with winter temperatures especially for PICE trees, i.e. potential photosynthetic activity at low temperatures in winter, where tissue formation is no

longer possible (Hoch et al., 2002 and references therein). Thereby, the concentration of NSC is increasing and is already available for tissue formation as soon as temperatures allow for it. Future studies on CC [%] on an annual and even intra-annual resolution could help improve our understanding of the influence of winter temperatures on the CC [%] in tree rings.

In PICE trees, the CC [%] is slowly and steadily increasing over time, potentially reacting more slowly to

increased temperatures, prolonged growing seasons and rising mean annual temperatures. In contrast, LADE series from modern trees exhibit a rapid increase over the past 30 years, obviously benefitting from increased

temperatures during the growing season. The CC [%] in Holocene wood remains and modern wood samples could therefore serve as a novel additional proxy in multi-proxy approaches, offering the opportunity to test the temperature sensitivity of diverse tree species, before combining them into chronologies. A combination of differently reacting tree species allows a complementation of temperature-CC [%] relationships, which may

result in more robust climate reconstructions.

Moreover, the PICE trees at two modern sampling sites showed highly significant correlation coefficients with sunshine duration especially during the winter season. As evergreen conifers are known to be photosynthetically active throughout the year as long as temperatures permit photosynthetic activity (Hoch et al., 2002 and references therein), PICE trees may potentially benefit from increased sunshine duration and be able to increase

the carbon acquisition during the winter season. Therefore, the CC [%] in PICE trees may even represent an annual signal. On the other hand, significant correlations found for sunshine duration and cloud cover with LADE CC [%] series may represent artefacts, as LADE trees lose their needles in autumn and carbon acquisition is only possible during the growing season.

As the framework of the project AHTRIR included both the analysis of living and subfossil wood, Holocene

wood remains were also investigated for signs of degradation. Most samples were well-preserved, and for the period from 9,000 to 3,500 years b2k, the CC [%] also varied between 30-40 CC%. Therefore, the CC [%] in living and subfossil wood samples is comparable. Only a small number of outliers was found (see also section 2.6), where CC [%] values showed pronounced decreases mostly appearing in the outermost rings as well as along cracks in the wooden material. We assume that these tree-ring sections have been affected by weathering

and therefore reveal a high degree of degradation, whereas the other rings have been well preserved (Fig. S6). Although the potential degradation of subfossil wood might have an impact, CC [%] of modern and subfossil wood is comparable despite a few outliers, which leads to the conclusion that long-term variations in Holocene CC [%] could serve as an indicator of climate variations. Moreover, there is no trend detected in CC [%] over time (i.e. towards the past) which would be expected in case degradation would have been a major driver of CC

[%] variations.

The low-frequency trends exhibited in the mean series of Holocene CC [%] in the period from 9000-3500 years b2k illustrate the potential of CC [%] as an additional proxy. The arithmetic mean CC [%] series shows pronounced decreases after known cold events in the Early and Mid-Holocene, whereas a continuous increase is observed between 7350 and 6250 years b2k, which could be the result of increased temperatures and more

favorable growing conditions for trees at the upper tree-line. However, the investigation of the individual species also illustrates differences in variations between LADE and PICE approving the observed differences in species within modern samples. A complete understanding of CC [%] variations in different tree species and the influence of environmental conditions on CC [%] will help to further improve the robustness of this novel proxy.

The presented study could benefit, but was at the same time limited by the framework of the project in which it

was performed: the vast advantage of the presented study are thousands of individual cellulose samples from both living and subfossil wood material distributed over large parts of the Holocene, which allowed the investigation of their CC [%] and served as a testbed for the temporal study of CC [%] in tree rings. However, we were at the same time limited by the high number of samples, which so far did not allow the analysis of replicates within this project. Further, the high-Alpine tree species used in this project often reveal very narrow

rings and the amount of extracted α-cellulose was just sufficient for further analysis. As the initial aims of the project did not include the closer analysis of CC [%] and its variation but was rather a concept that developed

during the progress of the project, the sampling and analysis of replicates has not been conducted so far. Yet, in a study performed earlier from the Lötschental in Switzerland, we evaluated the natural variability of CC [%] on different LADE tree-ring cores over time (Fig. S7, S8). It documents a mean standard deviation of 3.7 % in CC [%] for five individual cores from different trees of the same location. This standard deviation would even be

significantly smaller if the values of the different cores would be adjusted according to their mean values. Therefore, we are confident that replicates of LADE samples of the present study would be the same within a few couple of percent (approx. 3 to 4 %). In the current study, first measures to minimize and quantify the error of CC [%] have been presented; however, in future studies, it will be essential to accomplish a robust error estimation by a replicate sampling of the same tree.

**5 Conclusion and outlook**

For the first time, CC [%] and its temporal variability of the two coniferous Alpine tree-line species *Larix decidua* Mill. (European larch, deciduous) and *Pinus cembra* L. (Swiss stone pine, evergreen) have been investigated on a centennial scale for modern samples from living trees and on a multi-millennial scale for the Early and Mid-Holocene including the transition to the Late Holocene. The established CC [%] series revealed

species-specific long-term trends, independent of the location of the sampling site along the Swiss Alps in both past and present tree samples. First investigations of climate-cellulose relationships display the ability of CC [%] to record temperature at the Alpine tree-line.

As higher CC [%] values have been observed for low-elevation tree-ring material (unpublished data), the influence of elevation ought to be examined along an altitudinal gradient to quantify the decrease in α-cellulose

and thereby verify the hypothesis that not carbon, but temperature is the limiting factor resulting in limited sink activity which results in turn in lower CC [%] values. This goes along the line of non-structural carbon investigations.

In a succeeding study, the site- and species-sensitivity of multiple proxies (TRW, CC [%], $\delta D$, $\delta^{13}C$, $\delta^{18}O$) and the correlations between the individual proxies will be examined in relation to the environmental conditions per

site to quantify their ability for multi-millennial reconstructions based on a multi-proxy approach from tree rings (Ziehmer et al., in prep.). Thereby, the effects of variations in CC [%] on triple isotope records will be tested.

Moreover, CC [%] series ought to be established in annual resolution to establish a robust calibration and verification of the CC [%]-climate relationship as the reduced resolution of our samples (5-year tree-ring blocks) prohibits the proper analysis of the climatic response of cellulose and a robust calibration against instrumental

data. The existence of numberless CC [%] series in dendro-laboratories all over the world will allow a broad investigation of CC [%] in various species from wide-spread sampling sites in temporal resolution reaching from intra-annual to decadal tree-ring samples.

A future interlaboratory comparison comparable to the study by Boettger et al. (2007) could confirm the comparability of α-cellulose series between individual laboratories. Such a comparison should also include

investigations on the influence of preparation methods (milling vs. cutting wood), the individual steps and duration of the extraction, the role of the sample size, as well as the influence of tree species, juvenile vs. mature tree rings, and heartwood vs. sapwood. Further, purity of the extracted α-cellulose can be checked by FTIR spectra (Galia, 2015). Such an interlaboratory comparison is essential and prerequisite for the assessment of the accuracy of CC [%] and comparison of CC [%] series among different tree-ring laboratories.

Further, the analysis of a relationship between CC [%] and wood density would be highly interesting. These two proxies are usually not investigated within the same research project. However, establishing a link between these two variables might allow to draw conclusions on wood density by determining the CC [%] and vice versa. Hence, further research on the relationship between CC [%] and other tree-ring proxies (tree-ring width,

maximum latewood density, stable isotopes) is essential. Although the evaluation of CC [%] is obviously not easier than measuring tree-ring width, there is a significant potential of using it as an additional supplementary proxy, especially in those cases where CC [%] series are already existent in tree-ring laboratories and where climate is to be reconstructed in a multi-proxy approach.

A complete comprehension of environmental factors controlling the CC [%] in diverse tree species will result in
an improved understanding of physiological and biochemical processes in forming annual growth layers, and thus serve as an additional proxy in dendroclimatology. This study represents a first step into this direction and intends to motivate other tree-ring researchers in the field of dendroclimatology and stable isotope analysis to investigate their existing CC [%] series and to perform further research on these data.

**Data availability**

At present, data can be obtained upon request. As agreed among the project participants, datasets will be made available to the public after the official completion of the Alpine Holocene Tree Ring Isotope Records (AHTRIR) project.

**Author contribution**

M. M. Z. and M. L. designed the study. The subfossil wood sampling was mainly done by K. N and C. S
whereas the living wood sampling was performed by all authors. The preparation of the experiments as well as the statistical analysis and interpretation were carried out by M. M. Z. The drafting of the manuscript was accomplished by M. M. Z. with contributions from all co-authors.

**Competing interest**

The authors declare that they have no conflict of interest.

**Acknowledgements**

We thank A. Thurner and A. Oesterreicher from the Alpine Tree-Ring Group at University of Innsbruck for the sample preparation. Further thanks go to J. Lamprecht, Y. Rohrer, I. Zelenovic, R. Mülchi and M. Messmer for assistance during the sample preparation. Many thanks go to P. Nyfeler for technical support and experience.
The project is funded by the Swiss National Science Foundation (SNSF, 2000212_144255) and the Austrian
Science Fund (FWF, grant I-1183-N19) and is supported by the Oeschger Center for Climate Change Research, University of Bern, Bern, Switzerland (OCCR).

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

| site code | | FPCR | UAZR | VRR |
|---|---|---|---|---|
| sampling site | | Val d'Hérens, Mont Miné glacier | Haslital, Unteraar glacier, north shore Grimsel reservoir | Val Roseg |
| site character | site type | near timberline | near timberline | near timberline |
| | latitude °N | 46°04' N | 46°34' N | 46°26' N |
| | longitude °E | 7°33' E | 8°17' E | 9°51' E |
| | aspect | WSW | SSO | E |
| | elevation (m a.s.l.) | 1930-2000 | 1968-1986 | 2127-2190 |
| | number of trees | 4 | 4 | 8 |
| | period A.D. | 1680-2015 | 1810-2015 | 1630-2015 |
| | period A.D. with replication ≥ 4 | 1860-2015 | 1865-2015 | 1845-2015 (PICE) / 1865-2015 (LADE) |
| | mean segment length [years] | 208 | 173 | 209 (PICE) / 219 (LADE) |

**Table 1. Site characteristics of modern sites. Coordinates, aspect and elevation describe their geographical distribution from west to east. Four trees are sampled per site and species, respectively, resulting in eight trees cored at VRR, where both species are present at the tree-line. Site characteristics are complemented by the maximum growth period and the period covered by a sample replication of four trees in years A.D. as well as the mean segment length (MSL) in years.**

| site code | sampling site | site character | | | | |
|---|---|---|---|---|---|---|
| | | site type | latitude °N | longitude °E | aspect | elevation (m a.s.l.) |
| ahmo | Ahrntal, Moaralm | peat bog | 47°02' N | 12°05' E | SE | 1995 |
| ahst | Ahrntal, Starklalm | peat bog | 47°03' N | 12°07' E | S | 2080 |
| bih | Paznaun, Bielerhöhe | peat bog | 46°55' N | 10°06' E | N | 1930-2020 |
| eba | Ötztal, Ebenalm | lake/peat bog | 47°01' N | 10°57' E | NO | 2060-2170 |
| g | Ötztal, Gurgler Zirbenwald | peat bog | 46°51' N | 11°01' E | NW | 2060 |
| gdm | Kaunertal, Daunmoränensee | lake/peat bog | 46°53' N | 10°43' E | O | 2295 |
| ggua | Ötztal, Gurgler Alm | peat bog | 46°51' N | 11°00' E | W | 2150-2200 |
| gli | Kaunertal, Ombrometer | peat bog | 46°53' N | 10°43' E | NO | 2135-2160 |
| gp | Kaunertal, Gepatschferner | glacier foreland | 46°52' N | 10°44' E | W | 2060-2275 |
| hib | Defereggental, Hirschbichl | lake/peat bog | 46°54' N | 12°15' E | E | 2140 |
| kofl | Ahrntal, Kofler Alm | peat bog | 46°57' N | 12°06' E | S | 2165-2190 |
| mazb | Vinschgau, Marzoneralm/B | peat bog | 46°35' N | 10°57' E | N | 2125 |
| mazc | Vinschgau, Marzoneralm/C | peat bog | 46°35' N | 10°57' E | N | 2120 |
| maze | Vinschgau, Marzoneralm/E | peat bog | 46°35' N | 10°57' E | N | 2125 |
| mm | Val d'Hérens, Glacier du Mont Mine | glacier foreland | 46°02' N | 7°33' E | NNE | 1980-2010 |
| mort | Morteratschgletscher | glacier foreland | 46°25' N | 9°56' E | W | 2040-2050 |
| rt | Rojental | lake/peat bog | 46°48' N | 10°28' E | SO | 2400 |
| tah | Passeier, Timmeltal | peat bog | 46°54' N | 11°08' E | S | 1975-2260 |
| tsc | Val Roseg, Tschiervagletscher | glacier foreland | 46°24' N | 9°53' E | NW | 2115-2210 |
| ua | Haslital, Unteraargletscher | glacier foreland | 46°34' N | 8°13' E | E | 1950 |
| ulfi | Ultental, Fiechtsee | lake/peat bog | 46°28' N | 10°50' E | N | 2110 |
| uwba | Ultental, Weißbrunnalm | peat bog | 46°28' N | 10°49' E | NE | 2330 |
| zer | Mattertal, Zermatt, Findelengletscher | glacier foreland | 46°03' N | 7°47' E | N | 2300-2330 |

**Table 2. Site characteristics of Holocene sites. Given are the site codes as well as the geographic location expressed by latitude, longitude and aspect. Elevation of sampling sites is comparable with modern sampling sites, occasionally exceeding the current tree-line.**

| sample | species | start [years b2k (A.D.)] | end [years b2k (A.D.)] | years | PO | PO (5-year) | number of samples (5-year) |
|---|---|---|---|---|---|---|---|
| UAZR-1 | PICE | 135 (1865 A.D.) | - 14 (2014 AD) | 150 | 50 | 10 | 30 |
| UAZR-2 | PICE | 140 (1860 A.D.) | - 14 (2014 AD) | 155 | 82 | 17 | 31 |
| UAZR-3 | PICE | 165 (1835 A.D.) | - 14 (2014 AD) | 180 | 61 | 13 | 36 |
| UAZR-4 | PICE | 190 (1810 A.D.) | - 14 (2014 AD) | 205 | 32 | 7 | 41 |
| FPCR-1 | LADE | 320 (1680 A.D.) | - 14 (2014 AD) | 335 | 150 | 30 | 67 |
| FPCR-3 | LADE | 155 (1845 A.D.) | - 14 (2014 AD) | 170 | 30 | 6 | 34 |
| FPCR-5 | LADE | 155 (1845 A.D.) | - 14 (2014 AD) | 170 | 45 | 9 | 34 |
| FPCR-6 | LADE | 140 (1860 A.D.) | - 14 (2014 AD) | 155 | 55 | 11 | 31 |
| VRR-1 | PICE | 205 (1795 A.D.) | - 14 (2014 AD) | 220 | 62 | 13 | 44 |
| VRR-1-2 | PICE | 195 (1805 A.D.) | - 14 (2014 AD) | 205 | 77 | 15 | 41 |
| VRR-3 | PICE | 175 (1825 A.D.) | - 14 (2014 AD) | 190 | 12 | 3 | 38 |
| VRR-4 | PICE | 155 (1845 A.D.) | - 14 (2014 AD) | 170 | 49 | 10 | 34 |
| VRR-5 | PICE | 240 (1760 A.D.) | - 14 (2014 AD) | 255 | 13 | 3 | 51 |
| VRR-6 | LADE | 135 (1865 A.D.) | - 14 (2014 AD) | 150 | 6 | 2 | 30 |
| VRR-7 | LADE | 135 (1865 A.D.) | - 14 (2014 AD) | 150 | 13 | 3 | 30 |
| VRR-8 | LADE | 370 (1630 A.D.) | - 14 (2014 AD) | 385 | 27 | 6 | 77 |
| VRR-9 | LADE | 175 (1825 A.D.) | - 14 (2014 AD) | 190 | 25 | 5 | 38 |

**Table 3. Individual tree characteristics per modern sampling site. Given are the tree species, start and end date of the α-cellulose content series in years b2k and years A.D. in brackets, respectively. The length of the individual series is enlisted in years and the estimation of the pith offset (PO) in annual resolution and calculated as number of 5-year blocks. The number of 5-year cellulose blocks covering the length of the cellulose series is shown.**

| sample | bag incl. sample [mg] | emptied bag [mg] | expected sample weight [mg] | sample weight [mg] | loss [mg] | loss [%] |
|---|---|---|---|---|---|---|
| 1 | 138.250 | 126.552 | 11.698 | 11.473 | 0.225 | 1.9 |
| 2 | 138.112 | 128.480 | 9.632 | 9.307 | 0.325 | 3.4 |
| 3 | 129.500 | 116.654 | 12.846 | 12.293 | 0.553 | 4.3 |
| 4 | 147.350 | 134.918 | 12.432 | 11.980 | 0.452 | 3.6 |
| 5 | 129.060 | 118.279 | 10.781 | 10.405 | 0.376 | 3.5 |
| 6 | 133.301 | 123.375 | 9.926 | 9.731 | 0.195 | 2.0 |
| 7 | 147.471 | 132.364 | 15.107 | 14.758 | 0.349 | 2.3 |
| 8 | 136.276 | 123.247 | 13.029 | 12.486 | 0.543 | 4.2 |
| 9 | 147.790 | 134.199 | 13.591 | 13.011 | 0.580 | 4.3 |
| 10 | 136.433 | 125.743 | 10.690 | 10.341 | 0.349 | 3.3 |
| 11 | 132.212 | 121.153 | 11.059 | 10.708 | 0.351 | 3.2 |
| 12 | 133.778 | 119.853 | 13.925 | 13.553 | 0.372 | 2.7 |
| 13 | 145.170 | 133.002 | 12.168 | 11.667 | 0.501 | 4.1 |
| 14 | 140.314 | 128.114 | 12.200 | 11.804 | 0.396 | 3.2 |
| 15 | 118.253 | 108.009 | 10.244 | 10.211 | 0.033 | 0.3 |
| 16 | 127.516 | 115.089 | 12.427 | 12.004 | 0.423 | 3.4 |
| 17 | 124.031 | 111.314 | 12.717 | 11.733 | 0.984 | 7.7 |
| 18 | 123.409 | 113.444 | 9.965 | 9.704 | 0.261 | 2.6 |
| 19 | 127.960 | 119.475 | 8.485 | 8.228 | 0.257 | 3.0 |
| 20 | 154.157 | 136.382 | 17.775 | 17.348 | 0.427 | 2.4 |
| 21 | 138.214 | 126.419 | 11.795 | 11.321 | 0.474 | 4.0 |
| 22 | 114.291 | 101.147 | 13.144 | 12.823 | 0.321 | 2.4 |
| 23 | 136.587 | 125.258 | 11.329 | 11.305 | 0.024 | 0.2 |
| 24 | 121.803 | 107.166 | 14.637 | 14.383 | 0.254 | 1.7 |
| 25 | 130.931 | 116.707 | 14.224 | 13.969 | 0.255 | 1.8 |
| 26 | 138.189 | 126.357 | 11.832 | 11.503 | 0.329 | 2.8 |
| 27 | 128.379 | 114.667 | 13.712 | 13.430 | 0.282 | 2.1 |
| 28 | 132.578 | 120.533 | 12.045 | 11.402 | 0.643 | 5.3 |
| 29 | 131.704 | 119.411 | 12.293 | 11.726 | 0.567 | 4.6 |
| 30 | 113.811 | 102.525 | 11.286 | 10.972 | 0.314 | 2.8 |
| 31 | 120.800 | 109.288 | 11.512 | 11.310 | 0.202 | 1.8 |
| 32 | 116.480 | 102.682 | 13.798 | 13.321 | 0.477 | 3.5 |
| 33 | 125.114 | 112.714 | 12.400 | 12.036 | 0.364 | 2.9 |
| 34 | 119.056 | 107.090 | 11.966 | 11.530 | 0.436 | 3.6 |
| 35 | 121.489 | 109.201 | 12.288 | 11.951 | 0.337 | 2.7 |
| 36 | 106.796 | 92.591 | 14.205 | 13.770 | 0.435 | 3.1 |
| 37 | 126.301 | 111.446 | 14.855 | 14.389 | 0.466 | 3.1 |
| 38 | 127.248 | 113.117 | 14.131 | 13.642 | 0.489 | 3.5 |
| 39 | 156.524 | 143.516 | 13.008 | 12.529 | 0.479 | 3.7 |
| 40 | 130.215 | 121.632 | 8.583 | 8.347 | 0.236 | 2.7 |
| 41 | 144.788 | 140.079 | 4.709 | 4.362 | 0.347 | 7.4 |
| 42 | 123.144 | 109.257 | 13.887 | 13.681 | 0.206 | 1.5 |

**Table 4. Calculated sample loss during unpacking of extracted α-cellulose from ANKOM Filter Bags (Type F57) for 42 samples (sample tree MM-602, PICE). The expected sample weight is calculated by subtracting the scaled weight of the emptied filter bag from the weight of the filled bag. The difference between the expected weight of α-cellulose and the scaled weight of unpacked cellulose defines the loss, which is given here in [mg] and [%].**

| site | sample number | species | mean CC [%] | σ CC [%] | minimum CC [%] | maximum CC [%] | range CC [%] |
|------|---------------|---------|-------------|----------|----------------|----------------|--------------|
| UAZR | UAZR-1 | PICE | 33,2 | 3,9 | 26,7 | 42,6 | 15,9 |
|      | UAZR-2 | PICE | 31,6 | 3,4 | 24,8 | 36,8 | 12,0 |
|      | UAZR-3 | PICE | 38,6 | 1,6 | 34,8 | 42,8 | 8,0 |
|      | UAZR-4 | PICE | 39,7 | 2,5 | 34,3 | 46,4 | 12,1 |
| FPCR | FPCR-1 | LADE | 29,0 | 6,5 | 18,3 | 48,9 | 30,6 |
|      | FPCR-3 | LADE | 29,1 | 6,9 | 15,2 | 38,5 | 23,3 |
|      | FPCR-5 | LADE | 32,9 | 6,7 | 20,9 | 51,6 | 30,7 |
|      | FPCR-6 | LADE | 28,3 | 5,6 | 21,8 | 46,6 | 24,8 |
| VRR  | VRR-1 | PICE | 32,9 | 5,2 | 21,2 | 41,9 | 20,7 |
|      | VRR1-2 | PICE | 33,0 | 4,5 | 20,8 | 42,0 | 21,2 |
|      | VRR-3 | PICE | 34,3 | 3,2 | 29,0 | 42,0 | 13,0 |
|      | VRR-4 | PICE | 37,2 | 3,9 | 29,2 | 45,5 | 16,3 |
|      | VRR-5 | PICE | 35,4 | 3,3 | 30,2 | 45,7 | 15,5 |
|      | VRR-6 | LADE | 30,6 | 5,4 | 19,9 | 39,5 | 19,6 |
|      | VRR-7 | LADE | 36,0 | 4,4 | 30,4 | 44,5 | 14,1 |
|      | VRR-8 | LADE | 26,3 | 6,0 | 16,3 | 39,4 | 23,1 |
|      | VRR-9 | LADE | 32,5 | 5,5 | 26,4 | 46,3 | 19,9 |

**Table 5. General statistics of the individual α-cellulose content series per tree from the modern sites. Given are the arithmetic mean values, the standard deviation as well as maximum and minimum α-cellulose content values as well as the range per tree. To simplify the comparison between species, the species per site and tree individuum are enlisted once more.**

|         | UAZR-1 | UAZR-2 | UAZR-3 | UAZR-4 |
|---------|--------|--------|--------|--------|
| UAZR-1  | 1      |        |        |        |
| UAZR-2  | -0.10  | 1      |        |        |
| UAZR-3  | 0.35   | 0.36   | 1      |        |
| UAZR-4  | 0.08   | 0.32   | 0.64   | 1      |

|         | FPCR-1 | FPCR-3 | FPCR-5 | FPCR-6 |
|---------|--------|--------|--------|--------|
| FPCR-1  | 1      |        |        |        |
| FPCR-3  | 0.31   | 1      |        |        |
| FPCR-5  | 0.74   | 0.18   | 1      |        |
| FPCR-6  | 0.79   | 0.39   | 0.67   | 1      |

|         | VRR-1  | VRR-1-2 | VRR-3  | VRR-4  | VRR-5  |
|---------|--------|---------|--------|--------|--------|
| VRR-1   | 1      |         |        |        |        |
| VRR-1-2 | 0.75   | 1       |        |        |        |
| VRR-3   | 0.66   | 0.44    | 1      |        |        |
| VRR-4   | 0.75   | 0.62    | 0.63   | 1      |        |
| VRR-5   | 0.75   | 0.41    | 0.70   | 0.68   | 1      |

|         | VRR-6  | VRR-7  | VRR-8  | VRR-9  |
|---------|--------|--------|--------|--------|
| VRR-6   | 1      |        |        |        |
| VRR-7   | 0.78   | 1      |        |        |
| VRR-8   | 0.35   | 0.54   | 1      |        |
| VRR-9   | 0.55   | 0.61   | 0.60   | 1      |

**Table 6. Pearson's correlation coefficient calculated for the individual trees per site and species calculated in their common growth period (sample replication ≥ 4 trees). Bold numbers indicate that correlations coefficients are significant (p < 0.05).**

**Figures**

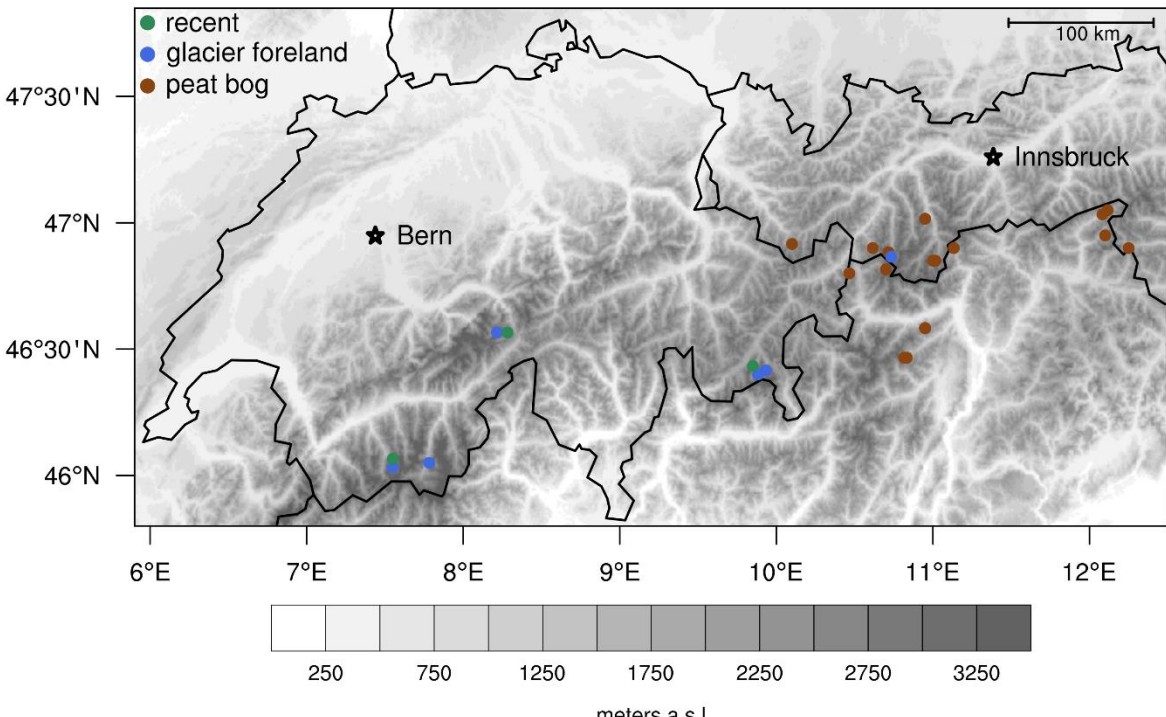

**Fig. 1. Sampling sites of Holocene wood remains and living tree individuals.** The sites are located along a SW-NE transect along the central European Alps and contain glacier forefields (blue), peat bogs and small lakes (maroon), where Holocene wood remains are collected, and modern sites, where living trees were cored (green). The modern sites are situated in the proximity of glacier forefield sites, thereby resembling similar growth conditions and serving as a connection between Holocene and present-day tree growth and environmental conditions.

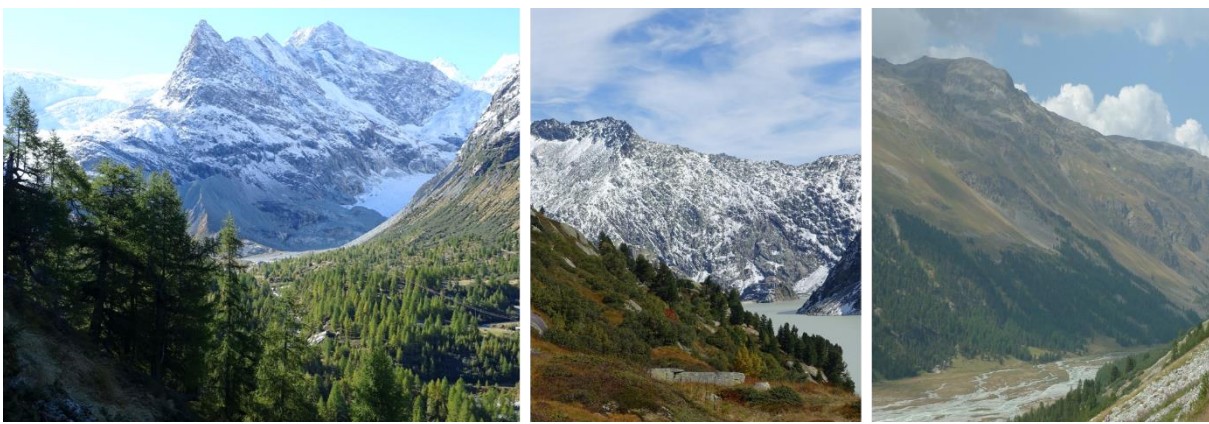

**Fig. 2. Location of modern sampling sites at the Alpine tree-line at Mont Miné glacier (FPCR, left), in front of Unteraar glacier (UAZR, center) and in the Val Roseg close to the Tschierva glacier (VRR, right). (Source: K. Nicolussi)**

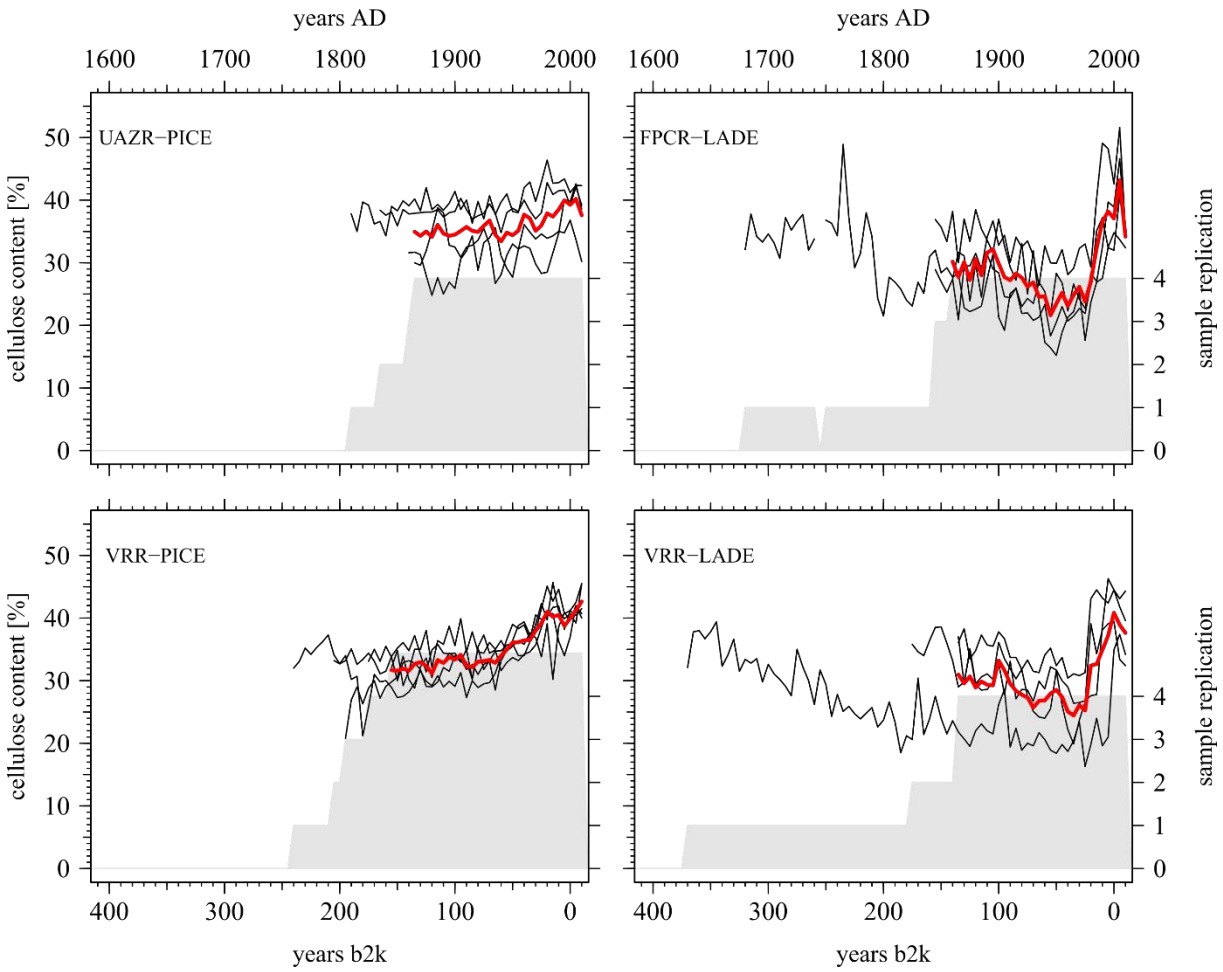

**Fig. 3. Individual cellulose content series for two coniferous species (LADE, PICE) at the modern sites. The individual cellulose content time series (black lines) are supplemented by the mean cellulose content per species and site (red line), calculated as the arithmetic mean of the individual trees for the common growth period (sample replication ≥ 4). The sample replication (grey area) displays the temporal coverage per site.**

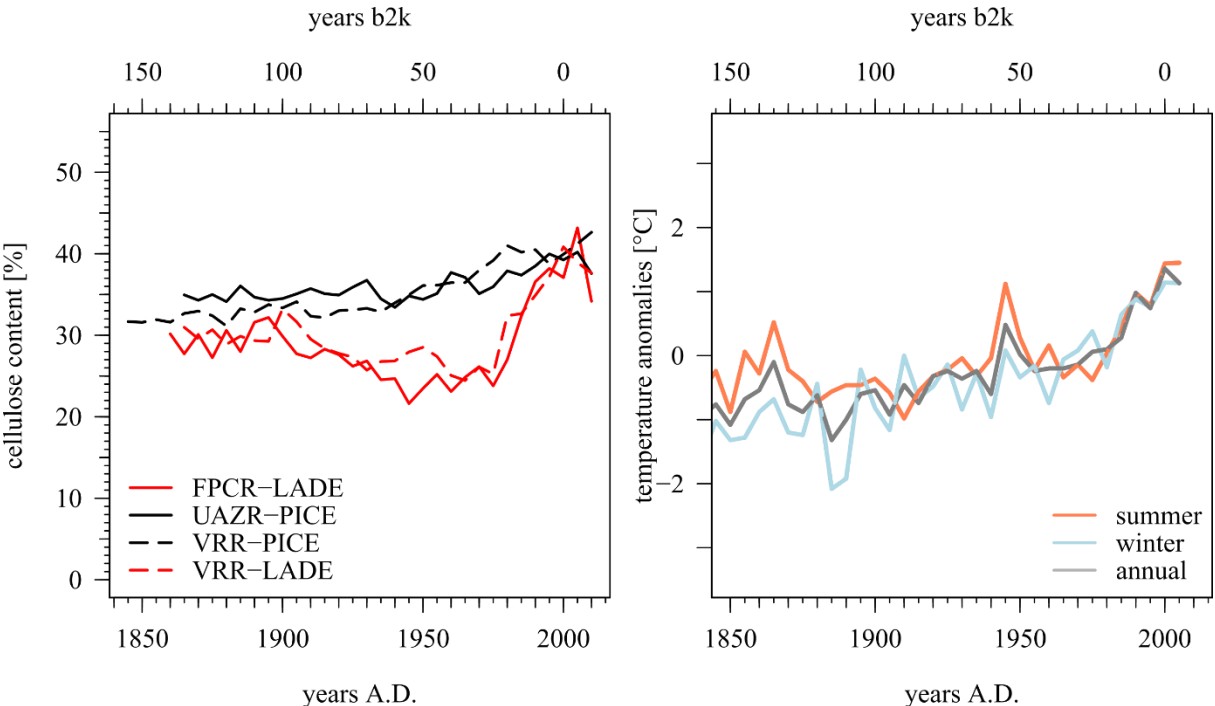

**Fig. 4. Modern cellulose content chronologies per site and species (left) and temperature anomalies shown for the NW and SW subregions within the Greater Alpine Region from HISTALP for summer (April-September) and winter (October-March) half-year and per year (right). The CC [%] chronologies are calculated as arithmetic means from individual series for their common period (sample replication ≥ 4). HISTALP data is given for summer season (April-September), winter season (October-March) and annual anomalies calculated as 5-year averages with respect to the sample resolution of CC [%] series. Temperature anomalies for SW and NW subregion in the displayed season means are equal.**

**Temperature**

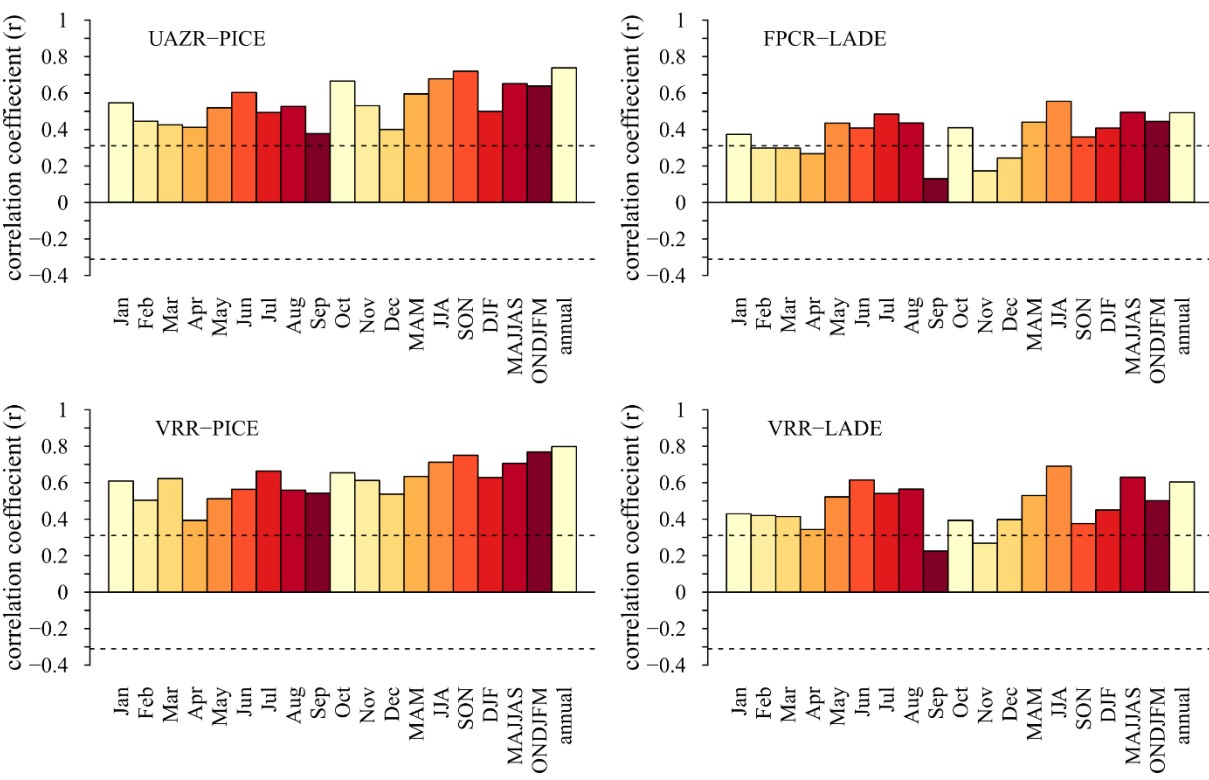

Fig. 5. Pearson's correlation coefficient for mean CC [%] and temperature anomalies (w.r.t. 1961-1990) for the period 1865 – 2005 A.D. UAZR and FPCR are correlated with HISTALP NW region data, whereas VRR species are correlated with the SW region dataset. The dashed horizontal lines indicate the level of significance ($p < 0.05$).

**Precipitation**

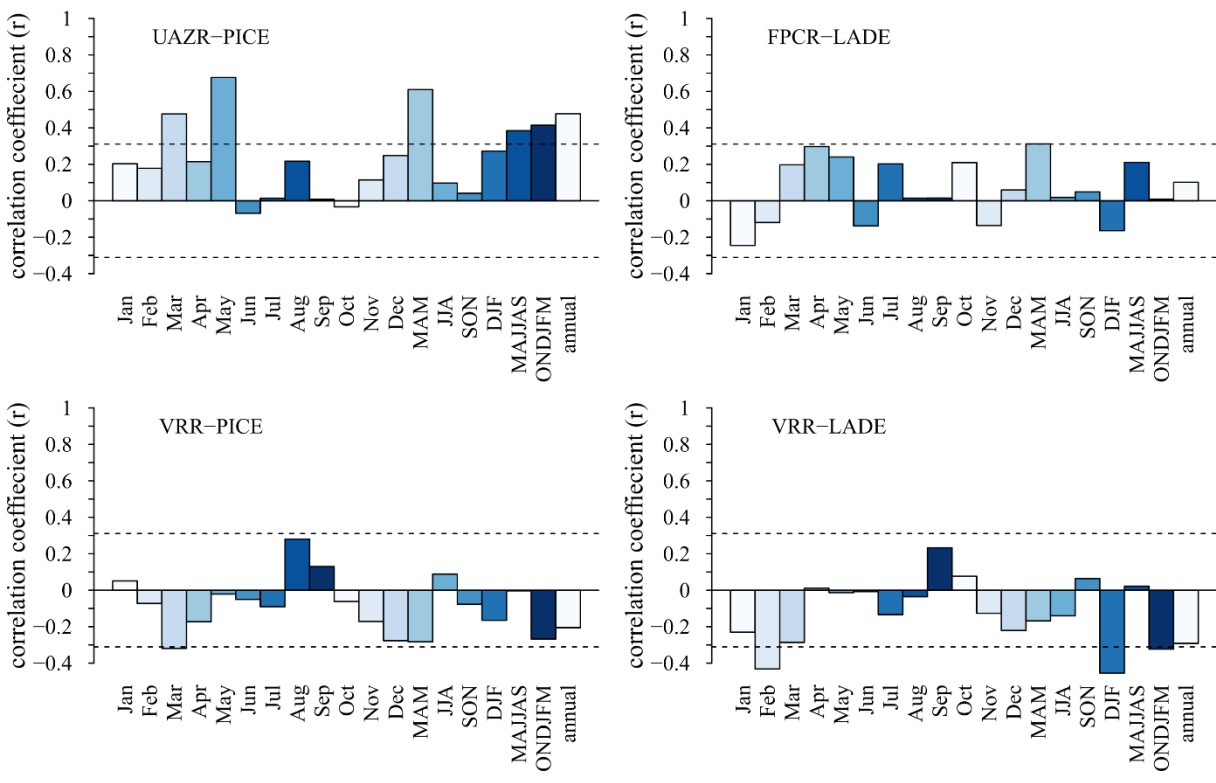

Fig. 6. Pearson's correlation coefficient for mean CC [%] and temperature anomalies (w.r.t. 1961-1990) for the period 1865 – 2005 A.D. UAZR and FPCR are correlated with HISTALP NW region data, whereas VRR species are correlated with the SW region dataset. The dashed horizontal lines indicate the level of significance ($p < 0.05$).

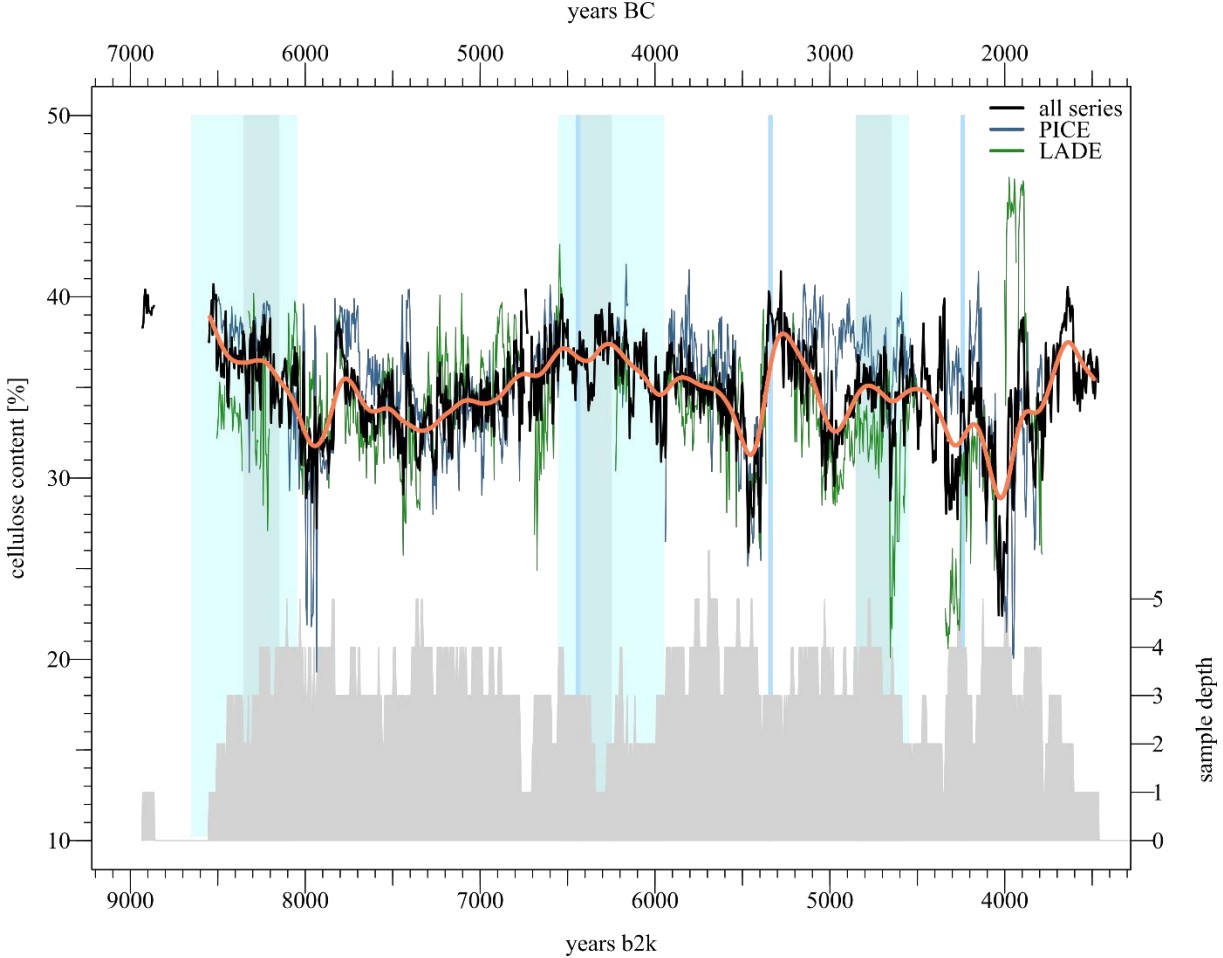

**Fig. 7. Progression of mean cellulose content during the time period 9000 to 3500 years b2k. Mean cellulose content is calculated for the two species *Larix decidua* Mill. (LADE, green line) and *Pinus cembra* L. (PICE, grey line); the overall mean series is shown in black and its smoothed values in orange. The sample depth is on the right side (grey polygon). Major cold phases are indicated as cyan rectangles, where darker colors implicit the cold events and the lighter colors their entire duration (Wanner et al., 2011). Additional cold events are marked as blue vertical lines (Wanner et al., 2015)**