# Peer review of "The potential of tree-ring cellulose content as a novel supplementary proxy in dendroclimatology"

_Biogeosciences, 2017_

## Referee Comment (RC1) · Anonymous Referee #1 · 16 May 2017

The authors have investigated if the cellulose content (CC) of tree-rings could be used as an additional palaeo-climatic proxy. The idea is indeed novel and the presented results are interesting. On the other hand, I have concerns regarding the reliability of the method.

- In my opinion and from my experience, I would expect the CC-values to be "operator-dependent", meaning that different laboratories would get to different results from the same tree-ring material. Various cellulose extraction protocols are used in different labs, see Boettger et al. 2007. An inter-lab comparison would therefore be crucial, in particular as there are no standards for verification. Further interfering methodological factors are also the preparation of the wood, e.g. milling (loss of powder during cellulose extraction) versus no milling (using slivers) and the recovery from filter/containers (also different methods are in use in different labs).

[Figure]

- Climate correlations with temperature may not be reliable as it seems to me that unrealistic significance levels have been used. The mean chronologies (Fig. 3) display a very high autocorrelation mainly because of the use of 5-yr blocked data. Therefore it is absolutely essential to correct the degree of freedom for autocorrelation, which seems not to have been done. This would increase the correlation coefficients needed to be significant. Obviously, increasing trends in the data will correlate with increasing temperature trends. Accordingly, even winter months appear to correlate significantly with CC, but I cannot believe that all months of the year influence CC as suggested by results in Fig. 5. Trends could be related to ageing, for instance, rather than climate.

- There are strong differences in the mean chronologies between Larix and Pinus for the same site (VEE), while chronologies from the same species but rather distant sites are similar. This points to the importance of biological factors rather than climate.

- The mechanisms resulting in varying CC are rather unclear. Some link to NSC and sink activity was proposed, but was not very understandable for me. The relationship between CC and wood density might be rather interesting to explore. Late-wood density is known as strong temperature indicator and it would be plausible to expect a relationship between CC-content and density.

- Due to the degradation of cellulose in old wood, the reliability of the subfossil CC series seems not so clear to me.

Overall, I find it worthwhile to investigate these data, but it seems premature to me to propose them as a palaeo-climate proxy.

---

## Referee Comment (RC2) · Anonymous Referee #2 · 30 Aug 2017

The paper by Ziehmer et al. highlights the possibility of using cellulose content in tree rings as a proxy for temperature. This paper is a rather technical paper that has two components: 1) a methodological aspect in which the authors discuss how to measure cellulose content in trees and 2) the application of using cellulose content as a proxy for temperature. While I believe that the approach of the authors is interesting and might even be promising, the authors have not convinced me of the accurate measurements of cellulose content. Lots of errors can be introduced in the method (which to a certain degree the authors discuss), but the paper lacks a clear estimation as to what the error on this method is. This could for example be accomplished by doing replicate sampling on the same tree. Another possibility is to split the paper in two papers, one which discusses the methodology and one which discusses the chronologies.

Major comments

While there are few grammatical and/or spelling mistakes, the paper should be improved for clarity. At times the paper is just very confusing. I suggest the authors try to shorten their paper and remove certain sections that make the paper unnecessary long and confusing (e.g. the discussion on whether to use dry weight before or after cutting, see more explanation below). In addition, the result section is also very confusing (see more details below)

Introduction

p2 L3-9: The authors argue that alpha-cellulose is the preferred substance for isotope analysis due to its long-term stability. I believe this is rather vague and the authors could give more details about the low mobility of cellulose, the fact that alpha cellulose is a singular chemical compound and the fact that it is also that the pathway from photosynthetic products to cellulose formation is more direct than the pathway to any of the other extractives (additional fractionations).

p2 L21-37: In this section, the authors discuss the fact that subfossil or fossil wood can have degradation of different wood components. The authors stress how this influences the isotope ratios and can have an effect on the ratios of the individual components. This is a major limitation of the study, but although the authors mention this, they don't seem to be worried that this might affect their study and there is no further mention of this in the rest of the paper and not even in the discussion.

Overall, the authors should bring in more discussion on the physiological aspects of the different components of wood formation in order to give the reader background into the possible limitations of the method. For example, cellulose/lignin/extractive ratios are known to differ between juvenile and mature wood and between heart wood and sapwood. It is also known to differ between normal wood and reaction wood (see for example Saka, 1991, Chemical composition and distribution, Ch 2 in Wood and Cellulosic Chemistry, Second Edition, Revised, and Expanded, as well as Rowell et

al 2012, Handbook of Wood Chemistry and Wood Composites (Second edition), CRC Press, London (2012), pp. 48-51)

The authors need to discuss this in the paper and need to address how this could affect their data.

In addition, the authors also should research additional papers studied on similar subjects. The following paper discusses lignin content as a proxy for temperature. Since lignin and cellulose are the two main components of wood, it seems logic that a change in one will also affect a change in the other. Gindl, W., Grabner, M. & Wimmer, R. 2000. The influence of temperature on latewood lignin content in treeline Norway spruce compared with maximum density and ring width.Trees 14: 409-414.

Results

P6, L5: The authors discuss that a determination of sample weight after cutting is essential. Considering that the study relies on cellulose content measurements, I believe that this is rather obvious. It is more logical to use dry weight after cutting rather than before cutting. I think it is a good idea of the authors to point it out and to discuss it, but I suggest the authors remove it from the methods (section 2.5). This will make that section much less confusing.

P6, L 11: The authors discuss the fact that a systematic error is introduced while the samples are unpacked. This is indeed a good addition, but the authors don't mention what they consider this error to be. Since it is a systematic error, the authors argue that the variability between samples should not be affected. However, the error means that small differences in cellulose content between samples cannot be interpreted. Therefore, it is very important that the authors discuss/estimate the error. Especially considering that they are looking at rather small differences in cellulose content. When looking at Table 4, it seems that the maximum weight loss during unpacking of the sample (after extraction) is 5.3 %. The authors could use this as a %error on their data. More accurately, the authors should determine the error by using replicate sampling of

the same tree.

P 6 section 3.1 and further: the use of % for cellulose content as well as % to express differences between sites is rather confusing and makes the paper difficult to read. Is there any way the authors can make this clearer? For example: p 6 L 22: UAZR1 and UAZR2 values are 10 percent lower than the other two trees. This could mean that their cellulose content drops from 40% to 30% (which is not the case), or that the cellulose content drops from 38 to 34 % (roughly 10% of 38% ∼4, so a 4 percent drop, which seems to be the correct interpretation here (?)). Another example: P6 L30: an increase in CC % over time by ∼5%. What does this mean? from 35 to 40 % or from 35 to 36.8% (reasoning that 5% of 35 is ∼1.3)

Discussion and Conclusion

The authors should revisit the methods used and include a discussion on the limitations of their method. Also, a discussion on the practical aspects of this method should be included: It is definitely not easier than measuring ring widths, so what is the advantage? Is there other information that has been revealed?

Minor comments

P4 section 2.5: this section is extremely confusing. If possible it should be rewritten P4 L32: I think the authors mean "1. dry weight" in the equation? P4, L33 "weighing" instead of weighting P5, L3: replace cellulose with sample P5, L19: add the . . .obtained in the form. . .

---

## Author Response (AR1)

**Reply to the review of Anonymous Referee #1**

The authors would like to thank anonymous referee #1 for the comments. In the following, referee's comments are given in bold, author's responses in plain text. Changes in the manuscript and suggested new text is quoted in red and italics together with page and line numbers of the revised and track-mode manuscript.

**The authors have investigated if the cellulose content (CC [%]) of tree-rings could be used as an additional palaeo-climatic proxy. The idea is indeed novel and the presented results are interesting. On the other hand, I have concerns regarding the reliability of the method.**

We appreciate that anonymous referee #1 recognizes the novelty of tree-ring cellulose content (CC [%]) as an additional supplementary proxy for paleo-climate reconstructions and that he evaluates our presented results as interesting. In the following, we would like to resolve his concerns regarding the reliability of our method by replying to referee #1's comments point by point.

**- In my opinion and from my experience, I would expect the CC-values to be "operatordependent", meaning that different laboratories would get to different results from the same tree-ring material. Various cellulose extraction protocols are used in different labs, see Boettger et al. 2007. An inter-lab comparison would therefore be crucial, in particular as there are no standards for verification. Further interfering methodological factors are also the preparation of the wood, e.g. milling (loss of powder during cellulose extraction) versus no milling (using slivers) and the recovery from filter/containers (also different methods are in use in different labs).**

Indeed, different tree-ring laboratories utilize different routine methods in order to produce tree-ring cellulose from wood material as shown in the interlaboratory comparison by Boettger et al. (2007) in the framework of the ISONET project. Boettger et al. (2007) described the extraction procedure in three main steps, namely (i) pretreatment, (ii) delignification and (iii) purification. Differences in the extraction procedure between the individual laboratories existed in the concentration of reagents, the treatment time as well as the temperatures (Boettger et al., 2007). The study showed that four out of nine laboratories produced holocellulose rather than α-cellulose, where holocellulose represents a combination of hemicelluloses and α-cellulose. Therefore, we agree that CC [%] values can be operator-dependent as the extraction method used in the individual laboratories determines if holocellulose or α-cellulose is extracted.

In our study, we focus on the investigation of α-cellulose which is defined as the part of cellulosic material being insoluble in 17% NaOH solution (Burton and Rasch, 1931; Cross and Bevan, 1912). In Boettger et al. (2007), five laboratories already used 17% NaOH solution in order to purify their samples. We would expect that α-CC [%] series among laboratories are comparable, whereas the holocellulose would reveal a positive offset compared to α-cellulose series, as the holocellulose series include hemicelluloses.

A future interlaboratory comparison comparable to the study by Boettger et al. (2007) could confirm the comparability of α-cellulose series between individual laboratories. Such a comparison should also include investigations on the influence of preparation methods (milling vs. cutting wood), the individual steps and duration of the extraction, the role of the sample size, as well as the influence of tree species, juvenile vs. mature tree rings, and heartwood vs. sapwood. Further, purity of the extracted

α-cellulose can be checked by FTIR spectra (Galia, 2015). Such an interlaboratory comparison is essential and prerequisite for the assessment of the accuracy of CC [%] and comparison of CC [%] series among different tree-ring laboratories.

However, the presented study is part of a project investigating Holocene climate variability in a multi-proxy approach, where we focused on the extraction of α-cellulose following a standardized extraction procedure as described in the methods section. Here, we tried to assess e.g. the loss during the recovery of cellulose from the filter bags of sliced samples (see results section) in order to estimate the error of our CC [%] time series.

The presented study benefits from the continuity of the applied methodology; however, we do agree that further investigation of the method by an interlaboratory comparison would be the essential next step in order to gain a better understanding of CC [%] series variations, its dependencies on parameters mentioned above and to allow comparisons of existing CC [%] series obtained from individual tree-ring laboratories.

The importance of a future interlaboratory comparison off CC [%] has been added to the conclusion and outlook of the manuscript (**p. 14, ll. 7-13**) and corresponds to our response to referee #1 made earlier in this reply:

*"A future interlaboratory comparison comparable to the study by Boettger et al. (2007) could confirm the comparability of α-cellulose series between individual laboratories. Such a comparison should also include investigations on the influence of preparation methods (milling vs. cutting wood), the individual steps and duration of the extraction, the role of the sample size, as well as the influence of tree species, juvenile vs. mature tree rings, and heartwood vs. sapwood. Further, purity of the extracted α-cellulose can be checked by FTIR spectra (Galia, 2015). Such an interlaboratory comparison is essential and prerequisite for the assessment of the accuracy of CC [%] and comparison of CC [%] series among different tree-ring laboratories."*

**- Climate correlations with temperature may not be reliable as it seems to me that unrealistic significance levels have been used. The mean chronologies (Fig. 3) display a very high autocorrelation mainly because of the use of 5-yr blocked data. Therefore, it is absolutely essential to correct the degree of freedom for autocorrelation, which seems not to have been done. This would increase the correlation coefficients needed to be significant. Obviously, increasing trends in the data will correlate with increasing temperature trends. Accordingly, even winter months appear to correlate significantly with CC, but I cannot believe that all months of the year influence CC as suggested by results in Fig. 5. Trends could be related to ageing, for instance, rather than climate.**

As described in section 2.7, we obtained meteorological data from the HISTALP database. For the individual climate parameters, we calculated monthly anomalies with the reference period 1961-1990. In order to obtain Pearson's correlation coefficients, five-year mean values were calculated for the individual climatic parameters, which matched the time step in our CC [%] series. We assume here that each data point (5-year blocked data) in the climate variable datasets and the CC [%] series is independent, since there is no data point overlap.

In order to clarify that data points in our correlation analysis are independent, the following sentence has been added to the manuscript (**p.6, ll. 22-23**):

*"We assume here that each data point (5-year blocked data) in the climate variable datasets and the CC [%] series is independent, since there is no data point overlap."*

We also investigated the possibility of age trends in our CC [%] series, although we did not expect to find any age trends as the modern CC [%] series of the two tree species did not reveal a common increase/ decrease over time (cf. p. 27, Fig. 4 in the manuscript). The age-alignment of all CC [%] series by their biological age, taking their estimated pith offset (PO) into account, illustrates that CC [%] series are not biased by age trends, which leads us to the conclusion that the trends which we find in CC [%] series are most probably driven by climatic variables (see Fig.1 at the end of the replies). The independence of the individual CC [%] samples and the lack of autocorrelation in the CC [%] series allow the presented calculation of Pearson's correlation coefficients between the 5-year CC [%] content series and the 5-year-average climate variable data sets.

The consideration of age trends in CC [%] series has been added to the discussion of the manuscript (**p.11, ll. 1-6**):

*"In order to exclude any age-related biases, possible age trends in CC [%] series have been investigated, even though they were not expected as the trends in modern CC [%] series of the two tree species diverge and do not reveal a common increase/ decrease over time. The age-alignment of both modern and Holocene CC [%] series by their biological age, taking their estimated pith offset (PO) into account, illustrates that CC [%] series are not biased by age trends, which leads us to the conclusion that the trends which we find in CC [%] series are most probably driven by climatic variables (Fig. S5).*

Regarding correlations with winter months: Indeed, especially the PICE trees show significant correlations also with winter temperatures. A potential explanation could be the fact that photosynthesis is still possible in winter (p. 10, ll.37-40). Thereby, the concentration of NSC is increasing and already available for tissue formation as soon as temperatures allow for it. Future studies on CC [%] on an annual and even intra-annual resolution could help improve our understanding of the influence of winter temperatures on the CC [%] in tree rings.

To clarify the link between correlations with winter months and CC [%], the following lines have been added to the discussion (**p. 12, ll. 4-9**):

*"This might explain the found correlations with winter temperatures especially for PICE trees, i.e. potential photosynthetic activity at low temperatures in winter, where tissue formation is no longer possible (Hoch et al., 2002 and references therein). Thereby, the concentration of NSC is increasing and is already available for tissue formation as soon as temperatures allow for it. Future studies on CC [%] on an annual and even intra-annual resolution could help improve our understanding of the influence of winter temperatures on the CC [%] in tree rings."*

**- There are strong differences in the mean chronologies between Larix and Pinus for the same site (VRR), while chronologies from the same species but rather distant sites are similar. This points to the importance of biological factors rather than climate.**

Indeed, the two different species *Larix decidua* Mill. and *Pinus cembra* L. at the sampling site in Val Roseg (VRR) exhibit different trends in their mean chronologies. As they both experience the same climate, the role of biological factors here is undoubted. This is kind of obvious, as we compare two coniferous species, where *Larix decidua* Mill. is a deciduous species, whereas *Pinus cembra* L. is an evergreen species; therefore, we would expect differences in their metabolism.

The fact that the individual CC [%] series from the same species at different sites are similar calls for a driving factor of regional extent, such as temperature.

In order to clarify the influence of biological factors and climate, the following section has been added to the discussion of the manuscript (**p. 11, ll. 7-16**), adapted from the reply to referee #1:

*"Further, the two different species Larix decidua Mill. and Pinus cembra L. at the sampling site Val Roseg (VRR) exhibit different trends in their mean chronologies. As they experience the same climate, the role of the biological factors here is undoubted. This is rather obvious as two coniferous species are compared, where Larix decidua Mill. is a deciduous species, whereas Pinus cembra L. is an evergreen species; therefore, differences in their metabolism are to be expected. Although the two species are both found at the upper tree-line and known to be adapted to the harsh environmental conditions, LADE is characterized as the light-demanding pioneer species which is often found in open settings, e.g. on glacier forefields, whereas PICE is, under undisturbed conditions, the highest rising species in the inner sections of the Alps and with that adapted to short vegetation periods (Ellenberg, 1996). The fact that individual CC [%] series from the same species at different sites are similar calls for a common driving factor of regional extent, such as temperature."*

**- The mechanisms resulting in varying CC are rather unclear. Some link to NSC and sink activity was proposed, but was not very understandable for me. The relationship between CC and wood density might be rather interesting to explore. Late-wood density is known as strong temperature indicator and it would be plausible to expect a relationship between CC-content and density.**

In this preliminary evaluation of CC [%], we investigated its variability, its link to climate and looked for potential mechanisms in a tree which could lead to variations in CC [%] and which are in turn dependent on climate. As we investigate two tree species growing at the Alpine tree-line, we know that their growth and therefore the cell formation is mostly restricted by temperature.

As discussed in the manuscript, research conducted by Körner and Hoch on the drivers of the climate-driven tree-line revealed that trees at the upper tree-line do not experience carbon shortage, but rather experience a lowered sink activity which results in a growth limitation (Hoch et al., 2002; Hoch and Körner, 2012, 2009; Körner, 1998). This simply means that a tree at the upper tree-line is still able to conduct photosynthesis, even though temperatures might be too low to allow tissue formation.

This represents the link to the variability in our CC [%]: short and cool growing seasons will lead to less tissue formation and a lower CC [%], whereas warm and prolonged growing seasons will lead to an increased tissue formation and therefore to a higher CC [%].

As the potential link between CC [%] and NSC is described in detail in the discussion section, no further changes have been made to this section.

Indeed, it would be highly interesting to analyse the relationship between CC [%] and wood density. Within the scope of this project, we tried to determine the density of our samples by Blue Intensity

(BI) measurements, as the conventional determination of density by X-ray images was not feasible due to the high amounts of individual samples (> 8,000 individual samples for the entire project). However, the determination of density by BI measurements in Holocene wood samples remains a challenge due to the different coloring of wood samples.

Still, future studies should further focus on the link between CC [%] and wood density and explore their relationship. Usually the determination of wood density and the determination of CC [%] for stable isotope analysis do not occur within the same research project. Establishing a link between these two variables might allow to draw conclusions on wood density by determining the CC [%] and vice versa. Hence, further research on the relationship between CC [%] and other tree-ring proxies (tree-ring width, maximum latewood density, stable isotopes) is essential.

The potential of investigating the link between CC [%] and wood density in future studies is shortly discussed in the conclusion and outlook section (**p. 14, ll. 14-21**):

*"Further, the analysis of a relationship between CC [%] and wood density would be highly interesting. These two proxies are usually not investigated within the same research project. However, establishing a link between these two variables might allow to draw conclusions on wood density by determining the CC [%] and vice versa. Hence, further research on the relationship between CC [%] and other tree-ring proxies (tree-ring width, maximum latewood density, stable isotopes) is essential. Although the evaluation of α-CC [%] is obviously not easier than measuring tree-ring width, there is a significant potential of using it as an additional supplementary proxy, especially in those cases where CC [%] series are already existent in tree-ring laboratories and where climate is to be reconstructed in a multi-proxy approach."*

**- Due to the degradation of cellulose in old wood, the reliability of the subfossil CC series seems not so clear to me.**

The investigation of CC [%] series has been conducted in the framework of the project *Alpine Holocene tree ring isotope records (AHTRIR)*, where the determination of CC [%] was initially used to determine the quality of the cellulose extraction before performing analysis of triple stable isotopes on the tree-ring cellulose. As the project aims at establishing triple isotope records for the past 9,000 years for the central European Alps, tree-ring material consists both of living wood material, but the largest part is based on findings of Holocene wood remains from glacier forefields, peat bogs and small lakes.

A short description of the project, in which the CC [%] study has been conducted, was added to the introduction of the manuscript (**p. 3, ll. 24-31**):

*"The current study is embedded in the project "Alpine Holocene Tree Ring Isotope Records (AHTRIR)" which aims at reconstructing Holocene climate variability using a multi-proxy approach for the past 9000 years. Therefore, tree-ring material consists of living wood material, but the largest part is based on findings of Holocene wood remains from glacier forefields, peat bogs and small lakes in the central European Alps. Initially, the determination of CC% was used to determine the quality of the cellulose extraction before performing analysis of triple isotopes ($\delta^2H$, $\delta^{18}O$, $\delta^{13}C$) on the tree-ring cellulose. However, it offers the unique opportunity to investigate the α-CC [%] and its variations in long-living trees from two high-Alpine coniferous tree species [...]"*

Although most samples were well preserved, we expected a certain grade of degradation in the Holocene wood remains. But for the period from 9,000 to 3,500 years b2k, we see that the CC [%] varies between 30-40 %, so the CC [%] of the subfossil samples is comparable to what we find in modern CC [%] series from living trees. Only a small number of outliers was found (as described in section 2.6), where CC [%] values showed pronounced decreases mostly appearing in the outermost rings as well as along cracks in the wooden material. We assume that these tree-ring sections have been affected by weathering and therefore reveal a high degree of degradation, whereas the other rings have been well preserved.

In order to clarify this issue and illustrate the occurrence of degradation in the outermost rings vs. well-preserved inner rings, a supplementary graph will be added (see a first example graph in Fig. 2 at the end of the replies).

The potential influence of degradation on CC [%] series from Holocene wood remains is discussed on **p. 12, ll. 26-37**:

*"As the framework of the project AHTRIR included both the analysis of living and subfossil wood, Holocene wood remains were also investigated for signs of degradation. Most samples were well preserved, and for the period from 9,000 to 3,500 years b2k, the CC [%] also varied between 30-40 CC%. Therefore, the CC [%] in living and subfossil wood samples is comparable. Only a small number of outliers was found (see also section 2.6), where CC [%] values showed pronounced decreases mostly appearing in the outermost rings as well as along cracks in the wooden material. We assume that these tree-ring sections have been affected by weathering and therefore reveal a high degree of degradation, whereas the other rings have been well preserved (Fig. S6). Although the potential degradation subfossil wood might have been a limitation of this study, CC [%] of modern and subfossil wood is comparable despite a few outliers, which leads to the conclusion that long-term variations in Holocene CC [%] could serve as an indicator of climate variations. Moreover, there is no trend detected in CC [%] over time (i.e. towards the past) which would be expected in case degradation would have been a major driver of CC [%] variations."*

**Overall, I find it worthwhile to investigate these data, but it seems premature to me to propose them as a palaeo-climate proxy.**

We agree with referee #1 that further tests are necessary to evaluate the potential of CC [%] in tree rings as a paleo-climate proxy. Therefore, we already suggested in the title of the paper that the current manuscript discusses the potential of tree-ring CC [%] as an additional supplementary proxy rather than stating that we found a new proxy. Yet, we further emphasize this point by changing the title to: "*Preliminary evaluation of the potential of tree-ring cellulose content as a novel supplementary proxy in dendroclimatology*".

The title has been changed to the proposed new title.

This highlights the importance that this potential has to be further explored and verified by interlaboratory comparisons and our study is a first step into this direction that intends to motivate other tree-ring researchers in the field of dendroclimatology and stable isotope analysis to investigate their existing CC [%] series and to perform further research on these data.

The importance of an interlaboratory comparison is highlighted in the conclusion and outlook section (**p. 14, ll. 7-13**, shown earlier in this reply to referee #1). Further, a sentence has been added on the intention and motivation of this paper to stimulate further research on tree-ring CC [%].

*"This study represents a first step into this direction and intends to motivate other tree-ring researchers in the field of dendroclimatology and stable isotope analysis to investigate their existing CC% series and to perform further research on these data."*

**Figures**

[Figure]

**Figure 1.** Cellulose content (CC [%]) in *Pinus cembra* L. (PICE) and *Larix decidua* Mill. (LADE) aligned according to their cambial age in years (pith offset estimation is considered here). Shown are the individual series in black and the mean in red, as well as the sample replication indicated by the grey area at the bottom of each graph.

[Figure]

**Figure 2.** Example of CC [%] variations and degradation in a *Larix decidua* Mill. tree (ULFI-47). The tree exhibits a long-term trend in its CC [%] series, followed by a rapid decrease of CC [%] in its outermost rings, which is attributed to degradation of CC [%] due to exposition to weathering. Still, most of the tree is well preserved and suitable for CC [%] analysis.

Reply to the review of Anonymous Referee #2

The authors would like to thank anonymous referee #2 for the comments. In the following, referee's comments are given in bold, author's responses in plain text.

**The paper by Ziehmer et al. highlights the possibility of using cellulose content in tree rings as a proxy for temperature. This paper is a rather technical paper that has two components: 1) a methodological aspect in which the authors discuss how to measure cellulose content in trees and 2) the application of using cellulose content as a proxy for temperature. While I believe that the approach of the authors is interesting and might even be promising, the authors have not convinced me of the accurate measurements of cellulose content. Lots of errors can be introduced in the method (which to a certain degree the authors discuss), but the paper lacks a clear estimation as to what the error on this method is. This could for example be accomplished by doing replicate sampling on the same tree. Another possibility is to split the paper in two papers, one which discusses the methodology and one which discusses the chronologies.**

We appreciate the review of the anonymous referee #2, who evaluates our approach as interesting and potentially promising. Still, referee #2 states concern e.g. about the accuracy of the measurements. In the following, we would like to reply and clarify mentioned issues.

At first, referee #2 divides the paper into two parts, namely a methodological and an application part. In contrast to referee #2, we do not see these two sections as separate and independent parts. The methodology to determine the cellulose content (CC [%]) of tree rings is a conventional method used in the field of dendroclimatology containing of three major steps: (i) wood preparation, (ii) cellulose extraction (in our case α-cellulose extraction) and (iii) the calculation of the CC [%] based on the wood and cellulose dry weight (cf. sections 2.3 – 2.5, pp. 4-5). As mentioned in the introduction of the manuscript (p. 2, ll. 17-20), the method of CC [%] determination is mostly used as a tool for determining the degradation state in subfossil wood and for evaluating the quality of the cellulose extraction. Therefore, the methodology itself is not novel; however, the application in form of CC [%] series which are investigated over time and the potential for an additional supplementary proxy in tree rings is indeed novel.

The current study has been developed in the framework of the project *Alpine Holocene Tree Ring Isotope Records (AHTRIR)*. The aim of the project is to develop triple tree-ring isotope records ($\delta^2$H, $\delta^{18}$O, $\delta^{13}$C) based on Holocene wood remains from glacier forefields, peat bogs and small lakes in the central European Alps to reconstruct climate by a multi-proxy approach for the past 9,000 years. Thereby, the framework of the project allowed the investigation of CC [%] series of both modern tree rings and subfossil wood remains and their variability over large parts of the Holocene in order to gain a better understanding of CC [%] in tree rings and its temporal variation.

*"The current study is embedded in the project "Alpine Holocene Tree Ring Isotope Records (AHTRIR)" which aims at reconstructing Holocene climate variability using a multi-proxy approach for the past 9000 years. Therefore, tree-ring material consists of living wood material, but the largest*

*part is based on findings of Holocene wood remains from glacier forefields, peat bogs and small lakes in the central European Alps. Initially, the determination of CC [%] was used to determine the quality of the cellulose extraction before performing analysis of triple isotopes ($\delta^2H$, $\delta^{18}O$, $\delta^{13}C$) on the tree-ring cellulose. However, it offers the unique opportunity to investigate the α-CC [%] and its variations in long-living trees from two high-Alpine coniferous tree species [...]"*

The presented study could benefit, but was also limited at the same time by the framework of the project: the vast advantage of the presented study are thousands of individual cellulose samples from both living and subfossil wood material distributed over large parts of the Holocene, which allowed the investigation of their CC [%] and served as a testbed for the temporal study of CC [%] in tree rings. However, we were at the same time limited by the high amount of samples, which so far did not allow the analysis of replicates within this project. Further, the high-Alpine tree species used in this project often reveal very narrow rings and the amount of extracted cellulose was just sufficient for further analysis. As the initial aims of the project did not include the closer analysis of CC [%] and its variation but was rather a concept that developed during the progress of the project, the sampling and analysis of replicates has not been conducted so far. Yet, in a study performed earlier from the Lötschental in Switzerland (unpublished measurements) we evaluated the natural variability of CC [%] on different larch tree-ring cores over time (see Fig. 1, 2 at the end of the replies). It documents a mean standard deviation of 3.7% in CC [%] for five individual cores from different trees of the same location. This standard deviation would even be significantly smaller when the values of the different cores would be adjusted according to their mean values. Therefore, we are confident that replications of larch samples of the present study would be the same within a few couple of percent (approx. 3 to 4 %).

Therefore, we do agree with referee #2 that for a robust error estimation a replicate sampling of the same tree would be preferential in the future. In the current study, we present first procedures to minimize and quantify the error, but we do agree that this is not yet complete and does not represent the accuracy of the method. Definitely, there is the need for another study on the influence and accuracy of the method.

A comment on the limitation in error estimation and replicate sampling has been added to the discussion of the manuscript (**p. 13, ll. 5-21**):

*"The presented study could benefit, but was at the same time limited by the framework of the project in which it was performed: the vast advantage of the presented study are thousands of individual cellulose samples from both living and subfossil wood material distributed over large parts of the Holocene, which allowed the investigation of their CC [%] and served as a testbed for the temporal study of CC [%] in tree rings. However, we were at the same time limited by the high amount of samples, which so far did not allow the analysis of replicates within this project. Further, the high-Alpine tree species used in this project often reveal very narrow rings and the amount of extracted cellulose was just sufficient for further analysis. As the initial aims of the project did not include the closer analysis of CC [%] and its variation but was rather a concept that developed during the progress of the project, the sampling and analysis of replicates has not been conducted so far. Yet, in a study performed earlier from the Lötschental in Switzerland (unpublished measurements) we evaluated the natural variability of CC [%] on different larch tree-ring cores over time (Fig. S7, S8). It documents a mean standard deviation of 3.7% in CC [%] for five individual cores from different trees of the same location. This standard deviation would even be significantly smaller when the values of the different cores would be adjusted according to their mean values. Therefore, we are*

*confident that replications of larch samples of the present study would be the same within a few couple of percent (approx. 3 to 4 %).*

*In the current study, first measures to minimize and quantify the error of CC [%] have been presented; however, in future studies, it will be essential to accomplish a robust error estimation by a replicate sampling of the same tree."*

Influences on CC [%] due to juvenile wood vs. mature wood, heart vs. sapwood, the influence of tree species and also the influence of preparation steps such as cutting vs. milling and the "storage" during extraction as well as the duration of the extraction need to be tested, as well as the chance of intercomparison between the individual laboratories. As suggested to reviewer #1, an intercomparison between those laboratories dealing with α-cellulose would be most suitable, as α-cellulose is well defined and its purity can be checked by FTIR determination (Galia, 2015).

The importance of a future interlaboratory comparison off CC [%] has been added to the conclusion and outlook of the manuscript (**p. 14, ll. 7-13**) and corresponds to our response to referee #1 made earlier in this reply:

*"A future interlaboratory comparison comparable to the study by Boettger et al. (2007) could confirm the comparability of α-cellulose series between individual laboratories. Such a comparison should also include investigations on the influence of preparation methods (milling vs. cutting wood), the individual steps and duration of the extraction, the role of the sample size, as well as the influence of tree species, juvenile vs. mature tree rings, and heartwood vs. sapwood. Further, purity of the extracted α-cellulose can be checked by FTIR spectra (Galia, 2015). Such an interlaboratory comparison is essential and prerequisite for the assessment of the accuracy of CC [%] and comparison of CC [%] series among different tree-ring laboratories."*

As the current study presents preliminary results on the analysis of CC [%] time series and the described methods simply summarize the methods used at the university of Bern, we would rather not split the paper, but present it as an initial work and inspiration for further studies on CC [%]. Further, replicate testing is not possible any longer due to finalization of the PhD of Malin Ziehmer, and we estimated errors as best as we could here, but we agree further studies are needed for the determination of the uncertainties associated with CC [%] time series.

**Major comments**

**While there are few grammatical and/or spelling mistakes, the paper should be improved for clarity. At times the paper is just very confusing. I suggest the authors try to shorten their paper and remove certain sections that make the paper unnecessary long and confusing (e.g. the discussion on whether to use dry weight before or after cutting, see more explanation below). In addition, the result section is also very confusing (see more details below)**

We accept that the section on the dry weight determination may appear confusing to the reader. The aim was to make the reader aware of the fact that there is a loss of sample material during the process of cutting (which is potentially analogue to e.g. milling), so that it is essential to determine the weight after cutting. However, we will rephrase the section in order to simplify and clarify it (see details below).

In order to clarify the paper and shorten it at the same time, section 2.5 has been rewritten (see details further below).

**Introduction**

**p2 L3-9: The authors argue that alpha-cellulose is the preferred substance for isotope analysis due to its long-term stability. I believe this is rather vague and the authors could give more details about the low mobility of cellulose, the fact that alpha cellulose is a singular chemical compound and the fact that it is also that the pathway from photosynthetic products to cellulose formation is more direct than the pathway to any of the other extractives (additional fractionations).**

In fact, there is potential here to elaborate more on the role of (α-)cellulose as preferred substance for isotope analysis and discuss this fact in more detail. For example, McCarroll and Loader (2004) have addressed three major reasons for the shift from the analysis of whole wood to α-cellulose in stable isotope analysis: (i) the unambiguous link of tree ring CC [%] to a specific growth period, (ii) the isolation of cellulose as a single chemical component, which reduces potential problems caused by varying cellulose:lignin ratios and (iii) the greater level of homogeneity achieved during the purification of α-cellulose. In addition, Boettger et al. (2007) conducted an interlaboratory comparison on methods of cellulose preparation, as cellulose is traditionally used for isotopic analysis, which is underpinned by the interlaboratory comparison among nine stable isotope laboratories in Europe.
Due to the well-established role of cellulose for stable isotope studies in the field of dendroclimatology, we did not consider it necessary to elaborate in more detail its characteristics and advantages for the fact that it has been done earlier in well-known tree-ring stable isotope publications (Boettger et al., 2007; McCarroll and Loader, 2004) and we tried to keep our introduction compact.
However, following the referee's suggestions, we highlight the preferred role of cellulose by adding relevant references (Borella et al., 1999, 1998, Loader et al., 2013, 2003; Treydte et al., 2007).

The relevant references elaborating on the preferred role of cellulose in dendroclimatology have been added to the manuscript (**p.2, ll. 3**):

*"[…] the most preferred component is α-cellulose as a single chemical component extracted from tree-rings (Borella et al., 1999, 1998, Loader et al., 2013, 2003; McCarroll and Loader, 2004; Treydte et al., 2007)."*

**p2 L21-37: In this section, the authors discuss the fact that subfossil or fossil wood can have degradation of different wood components. The authors stress how this influences the isotope ratios and can have an effect on the ratios of the individual components. This is a major limitation of the study, but although the authors mention this, they don't seem to be worried that this might affect their study and there is no further mention of this in the rest of the paper and not even in the discussion.**

As mentioned earlier in this reply, the presented study is part of the project *Alpine Holocene Tree Ring Isotope Records (AHTRIR)*, where most of the tree-ring material is derived from Holocene wood remains from glacier forefields, peat bogs and small lakes, and only a small part of samples consists of modern living wood. Most of the Holocene wood samples are well preserved, but a degradation of

samples cannot fully be excluded. This fact revealed the starting point for the investigation of CC [%] to see if modern and Holocene wood CC [%] are comparable.

In fact, the CC [%] in modern and Holocene wood samples is comparable; however, we found outliers in Holocene CC [%], where CC [%] showed pronounced decreases (cp. section 2.6 Outlier detection and correction). We could attribute these low CC [%] values to the outermost rings of Holocene wood remains, where e.g. the exposition to weathering within glaciers, peat bogs or lakes could have led to a higher degree of degradation (see Fig. 2 in Reply_Referee1).

We do agree that the use of subfossil wood might be a limitation of this study; however, at the same time we could show that CC [%] levels in subfossil and modern wood are comparable and concluded that we could use long-term variations in Holocene CC [%] as an indicator of climate variations.

We further agree that we should discuss the potential influence of degradation on the CC [%] time series in the discussion, and reflect to what extent the degradation of individual CC [%] series could affect the potential of CC [%] as a potential supplementary proxy. In this regard, it is important to note that we have not detected a trend of CC [%] over time (i.e. towards the past) which would be expected when degradation would be a major driver of the variations.

The limitation of using subfossil wood samples in the framework of the project AHTRIR and the influence of degradation on CC [%] are discussed on **p.12, ll. 26-37**:

*"As the framework of the project AHTRIR included both the analysis of living and subfossil wood, Holocene wood remains were also investigated for signs of degradation. Most samples were well preserved, and for the period from 9,000 to 3,500 years b2k, the CC [%] also varied between 30-40 CC [%]. Therefore, the CC [%] in living and subfossil wood samples is comparable. Only a small number of outliers was found (see section 2.6), where CC [%] values showed pronounced decreases mostly appearing in the outermost rings as well as along cracks in the wooden material. We assume that these tree-ring sections have been affected by weathering and therefore reveal a high degree of degradation, whereas the other rings have been well preserved (Fig. S6). Although the potential degradation subfossil wood might have been a limitation of this study, CC [%] of modern and subfossil wood was comparable despite a few outliers, which led to the conclusion that long-term variations in Holocene CC [%] could serve as an indicator of climate variations. Moreover, there were no trend detected in CC [%] over time (i.e. towards the past) which would be expected in case degradation would be a major driver of variations."*

**Overall, the authors should bring in more discussion on the physiological aspects of the different components of wood formation in order to give the reader background into the possible limitations of the method. For example, cellulose/lignin/extractive ratios are known to differ between juvenile and mature wood and between heartwood and sapwood. It is also known to differ between normal wood and reaction wood (see for example Saka, 1991, Chemical composition and distribution, Ch 2 in Wood and Cellulosic Chemistry, Second Edition, Revised, and Expanded, as well as Rowell et al 2012, Handbook of Wood Chemistry and Wood Composites (Second edition), CRC Press, London (2012), pp. 48-51)**
**The authors need to discuss this in the paper and need to address how this could affect their data.**

This is a valid point and actually highlights why we have concentrated on the extraction of one single chemical component, i.e. the α-cellulose for our main purpose: the isotope investigations. It allows us to circumvent the biases that potentially could result from a changing composition (cellulose/lignin)

when analyzing bulk wood since the different components exhibit significantly different isotope compositions (Borella et al., 1999, 1998).

As we focus on α-cellulose in this study, we highlighted its preferred role in dendroclimatology by commenting on its advantage of being a single chemical component (**p. 2, ll. 3-7**):

*"A major advantage of using α-cellulose in isotope research is the extraction of one single chemical component of a tree-ring. Thereby it allows researchers to circumvent the biases that potentially could result from a changing composition (i.e. in the cellulose/lignin ratio) when analysing bulk wood since different components exhibit significantly different isotope compositions (Borella et al., 1999, 1998)."*

Indeed, our work currently displays a lack on the discussion of physiological aspects of wood formation and their potential influence on the CC [%] series. Again, our study is here limited by the availability of the sampling material, which in our case consists mainly of Holocene wood remains from glaciers etc. as mentioned above. Therefore, we are also limited here in the exploration of the influence of juvenile vs. mature wood, or heart vs. sap wood. Still, these limitations should be mentioned and further explored in a future study, e.g. in the framework of an interlaboratory comparison.

As our study is limited by the availability of sampling material, physiological aspects of wood formation and their potential influence on CC [%] cannot be properly explored in the framework of this study and would be part of future interlaboratory comparison studies, which we shortly discuss in the conclusion and outlook section (**p. 14, ll. 7-13**):

*"A future interlaboratory comparison comparable to the study by Boettger et al. (2007) could confirm the comparability of α-cellulose series between individual laboratories. Such a comparison should also include investigations on the influence of preparation methods (milling vs. cutting wood), the individual steps and duration of the extraction, the role of the sample size, as well as the influence of tree species, juvenile vs. mature tree rings, and heartwood vs. sapwood. Further, purity of the extracted α-cellulose can be checked by FTIR spectra (Galia, 2015). Such an interlaboratory comparison is essential and prerequisite for the assessment of the accuracy of CC [%] and comparison of CC [%] series among different tree-ring laboratories."*

Regarding the use of reaction wood, we tried to avoid reaction wood; for modern trees, cores were taken in parallel to the slope, and for Holocene wood remains, stem discs were available, so reaction wood could mostly be identified and if possible avoided, as it is usually done in dendroclimatology. Similar to the potential degradation of wood, these physiological influences need to be further investigated and will shortly be addressed in this study.

This has been clarified in section 2.1 (**p. 4, ll. 12-20**):

*Cores are taken at breast height from three radial sections; two in parallel to the slope, and one up-slope, thereby avoiding any kind of compression wood on the down-slope side of the tree.*
*The Holocene wood remains stem from glacier forefields, peat bogs and small lakes, which have been continuously collected over the last two decades (Joerin et al., 2006, 2008; Nicolussi, et al., 2009; Nicolussi et al., 2005; Nicolussi and Patzelt, 2000b). Wood material of the EACC (Eastern Alpine Conifer Chronology) has been merged and updated with subfossil samples of the same species and*

*altitude collected by continued sampling of wood remains and stem discs in glacier forefields (Nicolussi and Schlüchter, 2012). Similar to modern wood samples, the use of compression wood in these Holocene samples was avoided during the further analysis.*

**In addition, the authors also should research additional papers studied on similar subjects. The following paper discusses lignin content as a proxy for temperature. Since lignin and cellulose are the two main components of wood, it seems logic that a change in one will also affect a change in the other. Gindl, W., Grabner, M. & Wimmer, R. 2000. The influence of temperature on latewood lignin content in treeline Norway spruce compared with maximum density and ring width.Trees 14: 409-414.**

In general, there is so far only little literature focusing on cellulose and lignin content in tree rings and their potential to reconstruct climate.

The results of the above-mentioned literature are indeed interesting; however, the analysis of lignin is only conducted over 10 consecutive years on modern wood samples. Further, the reconstruction of temperature based on lignin results in an autumn temperature reconstruction (Sept-Oct). Besides the fact that the reconstruction is very short, the authors describe the method as time-consuming (and potentially expensive?).

In contrast to the study of Gindl et al. (2000), the determination of CC [%] is somewhat a by-product when extracting α-cellulose for the analysis of stable isotopes in tree rings. The determination does neither add additional time consumption nor cost, but results in additional information on the individual tree rings. In our case, and due to the framework of the project, we worked with 5-year tree-ring blocks, in order to reduce cost and time to analyse Holocene climate variability within a feasible time (we are talking of thousands of measurements). Therefore, we create 5-year mean values and are able to investigate long-term trends in CC [%] which would not be possible by the method described by Gindl et al. (2000).

Another difference between the two components lignin and cellulose is the link to the growing season, where cellulose will be produced during most of the growing season, whereas lignin will be produced towards the end of the growing season. Therefore, cellulose will potentially incorporate a more homogenous temperature signal of large parts of the growing season (in particular for the evergreen pine trees growing at the tree-line for which photosynthesis is possible more or less throughout the year), whereas lignin will only record end of season temperature.

In general, further investigations are needed on how the ratios of the main components lignin, hemicelluloses and cellulose in a tree ring change and affect each other (cf. Borella et al., 1999, 1998 for isotope differences). A low lignin value could either result in an increased hemicellulose content or CC [%]. It would be worth to investigate these ratios in a tree ring also in relation to climatic factors.

The study by Gindl et al. (2000) establishes a relationship between a major tree ring component and climate; however, the time period they cover is rather short and the method rather expensive. Therefore, we added a short comment on this study in the introduction of our study in order to emphasize the advantages of using α-cellulose as an additional proxy (**p. 3, ll. 4-10**).

*"For instance, Gindl et al. (2000) discussed lignin content as a temperature proxy for the late growing season (Sept-Oct). As lignin and cellulose are among the major components of a tree ring, and changes in one component will affect the cellulose/lignin ratio, a link between CC [%] and temperature can be expected as well. However, the study of Gindl et al. (2000) covers only a short*

*time period of 10 years and the method used is rather time-consuming, whereas the production of α-CC [%] is often a by-product when extraction α-cellulose for stable isotope analysis in tree rings."*

**Results**

**P6, L5: The authors discuss that a determination of sample weight after cutting is essential. Considering that the study relies on cellulose content measurements, I believe that this is rather obvious. It is more logical to use dry weight after cutting rather than before cutting. I think it is a good idea of the authors to point it out and to discuss it, but I suggest the authors remove it from the methods (section 2.5). This will make that section much less confusing.**

We do agree that this is rather confusing. Therefore, we will rephrase section 2.5 and focus there only on the dry weight after cutting, and shortly discuss the loss during cutting in the results section (as currently done at the beginning of the results section).

Section 2.5 (**p.5, ll. 15-29**) has been rephrased as follows:

*"The CC [%] is calculated from the dry weight of wood and the cellulose weight of the sample:*

$$CC \ [\%] = \left( \frac{\alpha-cellulose \ weight}{wood \ dry \ weight} \right) \times 100\% \qquad (1)$$

*The wood dry weight is thereby defined as the sample weight of the individual wood sample after cutting, and the cellulose weight refers to the weight of the extracted α-cellulose after being removed from the ANKOM filter bag.*

*A remaining source of error is the collection of dry α-cellulose from the ANKOM filter bags. The described slicing of wood samples facilitates the removal of α-cellulose from filter bags compared to ground wood samples and thereby reduces the loss of α-cellulose material when being collected from the filter bags. In this study, the resulting loss is examined for 42 samples, where the filter bag including the α-cellulose and the emptied filter bag are scaled in addition to the α-cellulose weight to give an accurate estimate of the observed loss."*

The loss during cutting and the importance of determining the dry weight of the wood sample after cutting is shortly discussed at the beginning of the results section (**p. 6, ll. 28-33**):

*"The mean sample loss during the cutting process amounts to 2.6 ± 1.7 % of the dry weight of an individual sample, where mean loss values range from 0.3 % up to 11.1 % per tree (data not shown here). Besides, very few individual samples experienced losses between 10 % and 30 % caused by wood pieces bouncing off during cutting or loss of material due to powdery wood substance as a result of degradation. Hence, for the precise calculation of CC [%], a determination of sample weight after cutting or milling is essential."*

**P6, L 11: The authors discuss the fact that a systematic error is introduced while the samples are unpacked. This is indeed a good addition, but the authors don't mention what they consider this error to be. Since it is a systematic error, the authors argue that the variability between samples should not be affected. However, the error means that small differences in cellulose content between samples cannot be interpreted. Therefore, it is very important that the authors discuss/estimate the error. Especially considering that they are looking at rather small differences in cellulose content. When looking at Table 4, it seems that the maximum weight loss**

**during unpacking of the sample (after extraction) is 5.3 %. The authors could use this as a %error on their data. More accurately, the authors should determine the error by using replicate sampling of the same tree.**

When unpacking the cellulose from filter bags, there is always the risk that smallest cellulose fibers remain in the filter bag or fly off during the removal; therefore, we assume the error to be systematic (especially since the samples were unpacked by the same person). Here we tried to estimate the error by investigating 42 individual filter bags with samples from one tree. However, we do see that the loss per sample varies in the range from 0.2% up to 7.7% at maximum, which results in a mean loss of 3.2 ± 1.4% (percent of extracted cellulose weight and not dry weight) for these 42 cellulose samples. This relative uncertainty in cellulose weight transfers directly to the relative uncertainty of the CC [%] determination (relative uncertainties are additive with the uncertainty of the dry weight after cutting is negligible). Relevant for the CC [%] variation is the variation of the relative uncertainty (± 1.4%) and not the relative uncertainty itself (3.2%), which only yields a mean offset of the whole curve.

Although the relative uncertainty slightly limits the interpretation of small differences between the individual 5-year CC [%] samples, they do not limit the investigation of trends in CC [%] time series. For example, modern CC [%] agree in their trends, even though they might not perfectly agree in every single data point.

According to our reply to referee #2, an elaboration on the relative uncertainty of CC [%] series has been added to the beginning of the results section (**p. 6-7, ll. 33-14**):

*"Second, when unpacking the cellulose material from the filter bags, there is always a risk that smallest fibers remain in the filter bag or fly off during the removal; therefore, we assume this error to be systematic. In this study, we estimated this error by the investigation of 42 filter bags, which revealed a mean loss of α-cellulose of 0.378 ± 0.163 mg (3.2 ± 1.4 %) (Table 4). Therefore, the calculation of the CC [%] results in the minimum CC [%] of the sample, as losses of cellulose in the order of 3.2 ± 1.4 % of the cellulose weight are to be expected, which would result in slightly increased CC [%] value. However, as the unpacking is accomplished equally for all samples, a systematic error is assumed which affects the CC [%] calculation in the same manner. It can be assumed that the systematic error does not influence the variability of the individual CC [%] series. Further, the relative uncertainty in cellulose weight transfers directly to the relative uncertainty CC [%] determination, as relative uncertainties are additive, with the uncertainty of the dry weight after cutting being negligible. Relevant for the CC [%] variation is the variation of the relative uncertainty (±1.4 %), and not the relative uncertainty itself (3.2 %), which only yields a mean offset of the whole curve. Although the relative uncertainty slightly limits the interpretation of slight differences between the individual 5-year CC [%] samples, it does not limit the investigation of trends in CC [%] series. The improvement in the CC [%] calculation and the estimation of the relative error justify the use of the term "CC [%]" rather than "cellulose yield", as sources of error are reduced and well estimated, and resulting calculations are close to the true values."*

**P 6 section 3.1 and further: the use of % for cellulose content as well as % to express differences between sites is rather confusing and makes the paper difficult to read. Is there any way the authors can make this clearer? For example: p 6 L 22: UAZR1 and UAZR2 values are 10 percent lower than the other two trees. This could mean that their cellulose content drops from 40% to 30% (which is not the case), or that the cellulose content drops from 38 to 34 % (roughly 10% of 38% ~4, so a 4 percent drop, which seems to be the correct interpretation here (?)).**

**Another example: P6 L30: an increase in CC % over time by ~5%. What does this mean? from 35 to 40 % or from 35 to 36.8% (reasoning that 5% of 35 is ~1.3)**

Indeed, this could be made clearer by using CC% instead of %, e.g. p.6, ll. 23-25:

> *"These differences are a result of low minimum values for UAZR-1 and UAZR-2, which are up to 10 CC% lower than for the other two trees (Table 5)."*

Apart from the very first part of the results section (p.5, l. 34 – p.6, l.15), we always meant CC% when there is written % throughout the entire results and discussion section, so we are consistent here, but we agree that this is not obvious. Therefore, we would change all % into CC% in order to be correct and avoid any misunderstanding.

In order to clarify the use of units, CC [%] is given in the unit CC% (instead of % only). Absolute variations are given in CC% as well. This should avoid any confusion on absolute and relative changes which have occurred before by only using %. In the manuscript, the following sentence has been added to clarify the use of units (**p. 7, ll. 16-18**):

*"For a better understanding of CC [%] variations, modern CC [%] series are analysed in their temporal variability per species and site, where the absolute values and absolute variations in CC [%] are given in the unit CC%."*

**Discussion and Conclusion**
**The authors should revisit the methods used and include a discussion on the limitations of their method. Also, a discussion on the practical aspects of this method should be included: It is definitely not easier than measuring ring widths, so what is the advantage? Is there other information that has been revealed?**

We already agreed earlier in this reply, that limitations such as the use of subfossil wood, i.e. physiological aspects, as well as methodological aspects concerning the extraction and their potential influence on our dataset should shortly be discussed.
Obviously, the extraction of α-cellulose from tree rings is not easier than measuring tree-ring width, and usually α-cellulose is only extracted in the course of the stable isotope determination in tree rings. However, we see a significant potential that CC [%] series, which often already exist in many tree-ring laboratories in large quantities, could be used as an additional supplementary proxy. This is especially the case for multi-proxy studies with the aim to reconstruct climate, as the presented project AHTRIR, where there is potential to compare the variability of the CC [%] with other climate-dependent tree-ring proxies such as tree-ring width, density and isotopes.

The limitations of the study such as potential limitation by degradation of subfossil wood samples or the missing information from replicate measurements are elaborated towards the end of the discussion (**p.12-13. ll. 26-21**):

*"As the framework of the project AHTRIR included both the analysis of living and subfossil wood, Holocene wood remains were also investigated for signs of degradation. Most samples were well preserved, and for the period from 9,000 to 3,500 years b2k, the CC [%] also varied between 30-40 CC%. Therefore, the CC [%] in living and subfossil wood samples is comparable. Only a small number of outliers was found (see also section 2.6), where CC [%] values showed pronounced decreases mostly appearing in the outermost rings as well as along cracks in the wooden material. We*

*assume that these tree-ring sections have been affected by weathering and therefore reveal a high degree of degradation, whereas the other rings have been well preserved (Fig. S6). Although the potential degradation subfossil wood might have been a limitation of this study, CC [%] of modern and subfossil wood is comparable despite a few outliers, which leads to the conclusion that long-term variations in Holocene CC [%] could serve as an indicator of climate variations. Moreover, there is no trend detected in CC [%] over time (i.e. towards the past) which would be expected in case degradation would have been a major driver of CC [%] variations.*

*The low-frequency trends exhibited in the mean series of Holocene α-CC [%] in the period from 9000-3500 years b2k illustrate the potential of CC [%] as an additional proxy. The arithmetic mean CC [%] series shows pronounced decreases after known cold events in the Early and Mid-Holocene, whereas a continuous increase is observed between 7350 and 6250 years b2k, which could be the result of increased temperatures and more favorable growing conditions for trees at the upper tree-line. However, the investigation of the individual species also illustrates differences in variations between LADE and PICE approving the observed differences in species within modern samples. A complete understanding of CC [%] on tree species and environmental conditions will help to further improve the robustness of this novel proxy.*

*The presented study could benefit, but was at the same time limited by the framework of the project in which it was performed: the vast advantage of the presented study are thousands of individual cellulose samples from both living and subfossil wood material distributed over large parts of the Holocene, which allowed the investigation of their CC [%] and served as a testbed for the temporal study of CC [%] in tree rings. However, we were at the same time limited by the high amount of samples, which so far did not allow the analysis of replicates within this project. Further, the high-Alpine tree species used in this project often reveal very narrow rings and the amount of extracted cellulose was just sufficient for further analysis. As the initial aims of the project did not include the closer analysis of CC [%] and its variation but was rather a concept that developed during the progress of the project, the sampling and analysis of replicates has not been conducted so far. Yet, in a study performed earlier from the Lötschental in Switzerland (unpublished measurements) we evaluated the natural variability of CC [%] on different larch tree-ring cores over time (Fig. S7, S8). It documents a mean standard deviation of 3.7% in CC [%] for five individual cores from different trees of the same location. This standard deviation would even be significantly smaller when the values of the different cores would be adjusted according to their mean values. Therefore, we are confident that replications of larch samples of the present study would be the same within a few couple of percent (approx. 3 to 4 %).*

*In the current study, first measures to minimize and quantify the error of CC [%] have been presented; however, in future studies, it will be essential to accomplish a robust error estimation by a replicate sampling of the same tree."*

Further, the conclusion and outlook section has been complemented by comments on the potential of future research on CC [%] and the following sentence has been added (**p. 14, ll. 18-21**):

*"Although the evaluation of α-CC [%] is obviously not easier than measuring tree-ring width, there is a significant potential of using it as an additional supplementary proxy, especially in those cases where CC [%] series are already existent in tree-ring laboratories and where climate is to be reconstructed in a multi-proxy approach."*

**Minor comments**
**P4 section 2.5: this section is extremely confusing. If possible it should be rewritten.**

We agree that this section might be confusing due to the elaboration on the role of dry weight of wood before and after cutting. We will simplify this section by reducing the content simply to the dry weight after cutting, which will clarify the calculation of CC [%].

Section 2.5 has been rewritten (see comment earlier in this reply).

**P4 L32: I think the authors mean "1. dry weight" in the equation?**

No, here we presented the general formula without focusing on 1. or 2. dry weight. However, being rewritten, we will eliminate this issue and define dry weight as the dry weight after cutting of the wood sample.

This has been eliminated as the entire section has been rewritten.

**P4, L33 "weighing" instead of weighting**

Indeed, this is a spelling mistake and will be corrected.

Due to rewriting section 2.5, this spelling mistake is no longer existing.

**P5, L3: replace cellulose with sample**

We would rather replace cellulose by cellulose material than by sample.

This has been corrected and reads now as follows:

"[…] thereby reduces the loss of cellulose material when being collected from the filter bags."

**P5, L19: add the . . .obtained in the form. . .**
Indeed, the article is missing and will be filled in.

This has been corrected and reads now as follows:

*"Data is obtained in the form of Coarse Resolution Subregional Means (CRSM) […]"*

**Figures**

[Figure]

**Figure 1.** Variability of CC [%] in larch tree rings from Lötschental (CH). The numbers correspond to tree cores from different trees.

[Figure]

**Figure 2.** Temporal variability of CC [%] in larch tree ring series from Lötschental (CH). The numbers correspond to tree cores from different trees.

**References**

[revised manuscript text omitted]